# Oncogenic BRAF disrupts thyroid morphogenesis and function via twist expression

Viviana Anelli[1], Jacques A Villefranc[1], Sagar Chhangawala[1],
Raul Martinez-McFaline[1], Eleonora Riva[2], Anvy Nguyen[1], Akanksha Verma[3,4],
Rohan Bareja[3,4], Zhengming Chen[5], Theresa Scognamiglio[6], Olivier Elemento[3,4],
Yariv Houvras[1,7,8]*

[1]Department of Surgery, Weill Cornell Medical College, New York Presbyterian Hospital, New York City, United States; [2]Section of Endocrinology, Department of Medical Science, University of Ferrara, Ferrara, Italy; [3]Institute for Computational Biomedicine, Weill Cornell Medical College, New York City, United States; [4]Department of Physiology and Biophysics, Weill Cornell Medical College, New York City, United States; [5]Department of Healthcare Policy & Research, Weill Cornell Medical College, New York City, United States; [6]Department of Pathology and Laboratory Medicine, Weill Cornell Medical College, New York Presbyterian Hospital, New York City, United States; [7]Sandra and Edward Meyer Cancer Center, Weill Cornell Medical College, New York Presbyterian Hospital, New York City, United States; [8]Department of Medicine, Weill Cornell Medical College, New York Presbyterian Hospital, New York City, United States

*For correspondence: yah9014@
med.cornell.edu

**Competing interests:** The authors declare that no competing interests exist.

**Abstract** Thyroid cancer is common, yet the sequence of alterations that promote tumor formation are incompletely understood. Here, we describe a novel model of thyroid carcinoma in zebrafish that reveals temporal changes due to BRAF$^{V600E}$. Through the use of real-time in vivo imaging, we observe disruption in thyroid follicle structure that occurs early in thyroid development. Combinatorial treatment using BRAF and MEK inhibitors reversed the developmental effects induced by BRAF$^{V600E}$. Adult zebrafish expressing BRAF$^{V600E}$ in thyrocytes developed invasive carcinoma. We identified a gene expression signature from zebrafish thyroid cancer that is predictive of disease-free survival in patients with papillary thyroid cancer. Gene expression studies nominated TWIST2 as a key effector downstream of BRAF. Using CRISPR/Cas9 to genetically inactivate a TWIST2 orthologue, we suppressed the effects of BRAF$^{V600E}$ and restored thyroid morphology and hormone synthesis. These data suggest that expression of TWIST2 plays a role in an early step of BRAF$^{V600E}$-mediated transformation.

## Introduction

Papillary and follicular thyroid cancers are highly prevalent malignancies arising from transformation of epithelial cells. Mutations in key signaling effectors of the mitogen activated protein kinase (MAPK) pathway, including BRAF and RAS, are found in 70% of thyroid cancers. A recurrent activating mutation in *BRAF, BRAF$^{V600E}$*, is found in 45% of papillary thyroid cancers (PTC), the most common histologic subtype (*Xing, 2005*). Although patients with PTC may be cured with surgery and radioiodine treatment, mutations in BRAF are associated with an increased risk of disease recurrence

(*Xing et al., 2015*). Therefore, it is important to understand the cellular and molecular mechanisms in thyrocytes that lead to malignant transformation by BRAF$^{V600E}$.

Mutations in BRAF have been linked to several mechanisms of malignant transformation. Expression of BRAF$^{V600E}$ has been demonstrated to increase thyrocyte migration and invasion through induction of an epithelial to mesenchymal transition (EMT) in vitro (*Baquero et al., 2013*). Transgenic mouse models demonstrate that expression of BRAF$^{V600E}$ leads to aggressive papillary thyroid carcinomas that progresses to poorly differentiated cancers and demonstrate a loss of sodium iodide symporter expression and a failure to concentrate iodine (*Knauf et al., 2005*, *2011*; *Chakravarty et al., 2011*). Yet many human BRAF$^{V600E}$ mutant PTCs are slow growing cancers, that may be clinically stable for years, and identifying these tumors is a key clinical challenge (*Haser et al., 2016a*, *2016b*). While current animal models largely recapitulate aggressive thyroid cancers, there is a need to identify the molecular characteristics that differentiate indolent thyroid cancer from more aggressive subtypes and understand the molecular mechanisms that are involved in progression.

Understanding the temporal consequences of BRAF$^{V600E}$ expression in thyrocytes and thyroid follicles is a key to deciphering the mechanism of malignant transformation. For this reason, we developed a zebrafish model to visualize the consequences of BRAF$^{V600E}$ expression on thyrocyte follicle structure, hormone synthesis, and organ morphogenesis. Expression of BRAF$^{V600E}$ in zebrafish thyrocytes leads to profound disruption of follicle structure and thyroid hormone production, changes that precede an increase in proliferation. Transgenic zebrafish that express BRAF$^{V600E}$ in thyrocytes develop thyroid carcinomas by one year of age with histopathologic hallmarks of human papillary thyroid cancer. Tumors from zebrafish harbor a gene expression signature that stratifies disease recurrence in patients with papillary thyroid carcinoma. We identify an orthologue of human TWIST2, *twist3*, as a key mediator of BRAF$^{V600E}$ induced EMT in thyrocytes. Using CRISPR/Cas9 gene editing we demonstrate that *twist3* loss of function suppresses the effects of BRAF$^{V600E}$ on follicle morphogenesis and hormone production. These studies provide critical insight into the earliest effects of oncogenic BRAF$^{V600E}$ in thyrocytes.

## Results

### Characterization of transgenic zebrafish expressing BRAF$^{V600E}$ in thyrocytes

In order to investigate the temporal consequences of oncogenic BRAF expression in thyrocytes, we created stable transgenic lines expressing either human BRAF$^{V600E}$ and a TdTomato reporter gene (tg-BRAF$^{V600E}$-TOM) or TdTomato alone (tg-TOM), both under control of a thyroid-specific promoter (*McMenamin et al., 2014*) (*Figure 1A*). At five days post-fertilization (dpf) control, tg-TOM larvae formed distinct well-organized thyroid follicles composed of TdTomato+ thyrocytes surrounding colloid containing thyroid hormone, positioned in the ventral aspect of the jaw (*Figure 1B*), as previously reported (*Wendl et al., 2002*; *Opitz et al., 2013*). In contrast, tg-BRAF$^{V600E}$-TOM larvae exhibited profound defects in thyroid follicle morphogenesis, forming disorganized clusters of thyrocytes (*Figure 1C*). This phenotype was followed by live imaging and was highly penetrant.

To investigate whether expression of BRAF$^{V600E}$ in thyroid follicular cells caused variation in thyroid hormone (T4, thyroxine) production or follicle number, we performed whole mount anti-T4 immunostaining to identify individual T4+ follicles. As expected, at 5 dpf tg-TOM larvae formed an average of five T4+ follicles along the ventral-medial axis (*Figure 1D and F*). In contrast, tg-BRAF$^{V600E}$-TOM larvae displayed a significant decrease in the number and size of T4+ follicles (*Figure 1E–G*). To examine the effects on proliferation in BRAF$^{V600E}$ thyrocytes, we stained tg-BRAF$^{V600E}$-TOM and tg-TOM with Sytox Green Nucleic Acid Stain and counted the total number thyrocytes at 5 dpf. Total thyrocyte number was comparable between tg-BRAF$^{V600E}$-TOM and tg-TOM control larvae (*Figure 1—figure supplement 1A–C*). Thyrocyte proliferation was further evaluated by BrdU incorporation into newly synthesized DNA. BrdU-positive TdTomato-positive cell number was not significantly different between tg-BRAF$^{V600E}$-TOM and tg-TOM control larvae (*Figure 1—figure supplement 1D–F*). Additionally, the total area and volume encompassed by TdTomato positive thyroid cells was comparable between tg-BRAF$^{V600E}$-TOM and control larvae (*Figure 1—figure supplement 1G and H*). These results reveal that expression of BRAF$^{V600E}$ in follicular

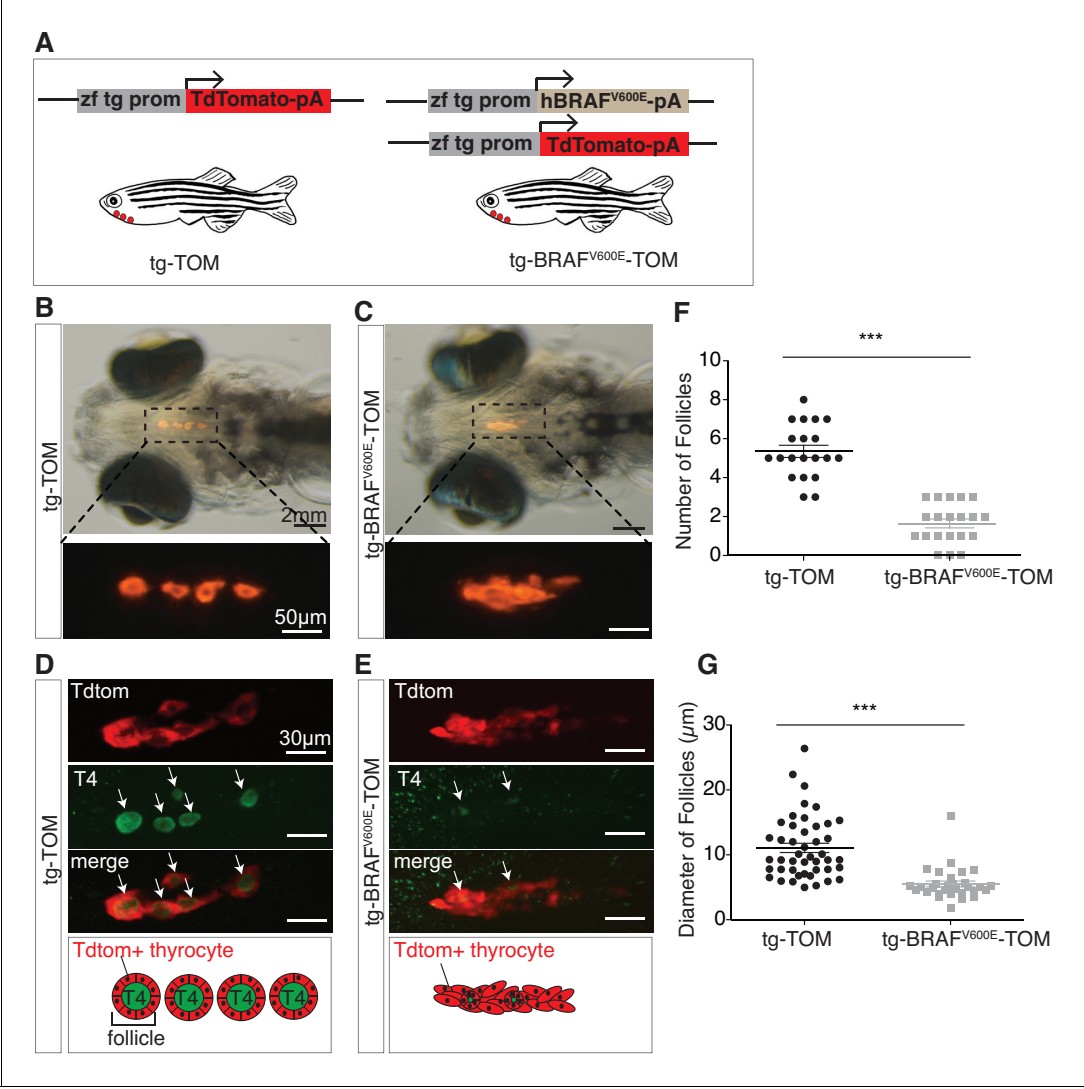

**Figure 1.** BRAF[V600E] expression in thyrocytes disrupts follicle structure. (**A**) Schematic diagram of the constructs used to create transgenic lines, containing a specific zebrafish thyroglobulin promoter (zf tg prom) driving expression of either human BRAF[V600E] or TdTomato fluorophore. (**B–C**) Brightfield micrographs overlayed on to an epifluorescent micrograph of a live 5 dpf tg-TOM and tg-BRAF[V600E]-TOM larvae. (**D–E**) Representative confocal images of 5 dpf fixed tg-TOM and tg-BRAF[V600E]-TOM larvae immunostained with anti-T4 antibody; white arrows indicate T4-positive follicles. (**F**) Quantification of the total follicle number in tg-TOM and tg-BRAF[V600E]-TOM larvae (n = 20). (**G**) Measurement of follicle diameter in tg-TOM and tg-BRAF[V600E]-TOM (n = 45 and n = 30, respectively). ***p≤0.001, Student's two-tailed *t*-test.

The following figure supplement is available for figure 1:

**Figure supplement 1.** Expression of BRAF[V600E] does not affect proliferation of larval thyrocytes.

cells disrupts follicle integrity, resulting in decreased number and size of follicles without altering cell number.

## Combined inhibition of BRAF[V600E] and MEK restores follicular morphology in thyrocytes expressing BRAF[V600E]

To investigate whether aberrant thyroid follicle morphogenesis in BRAF[V600E] larvae could be suppressed by inhibition of MAPK signaling, we treated tg-BRAF[V600E]-TOM and tg-TOM embryos with either dabrafenib, a specific inhibitor BRAF[V600E] (*Kainthla et al., 2014*) or selumetinib (*Kim and Patel, 2014*), a MEK inhibitor, or the combination of both inhibitors and observed thyroid structure

by live imaging. To limit defects caused by inhibition of MAPK signaling on early development, embryos were treated at doses compatible with normal development (*Figure 2A*). Vehicle (DMSO) treated tg-BRAF$^{V600E}$-TOM larvae exhibited disorganized follicular structure in contrast to tg-TOM larvae at 5 dpf (*Figure 2B*). Treatment with either dabrafenib or selumetinib alone was not sufficient to restore normal thyroid morphogenesis in tg-BRAF$^{V600E}$-TOM larvae. By contrast, combined treatment with both dabrafenib and selumetinib restored normal thyroid follicular structure (*Figure 2B*), T4 production (*Figure 2C*), and led to a significant increase in the number and size of follicles (*Figure 2D and E*). Importantly, inhibition of MAPK signaling in tg-TOM larvae had no overt effects on embryo development or thyroid follicular structure, but was associated with a modest decrease in follicle number, consistent with a proliferative requirement in follicle morphogenesis. Taken together, these results suggest that combined inhibition of MEK and BRAF$^{V600E}$ activity is required to restore normal follicular architecture and thyroid hormone (T4) production in tg-BRAF$^{V600E}$-TOM larvae.

## BRAF$^{V600E}$ expression leads to TGF-β and EMT gene expression changes in thyrocytes

To dissect the molecular pathways involved in the BRAF$^{V600E}$ follicular phenotype, we performed flow sorting of TdTomato-positive thyrocytes from tg-TOM or tg-BRAF$^{V600E}$-TOM transgenic larvae at 5 dpf and isolated RNA for next-generation sequencing. TdTomato-positive cells accounted for 0.076% and 0.060% of the total cells sorted in tg-TOM and tg-BRAF$^{V600E}$-TOM respectively (*Figure 3A*), further confirming that the absolute number of thyrocytes is comparable between

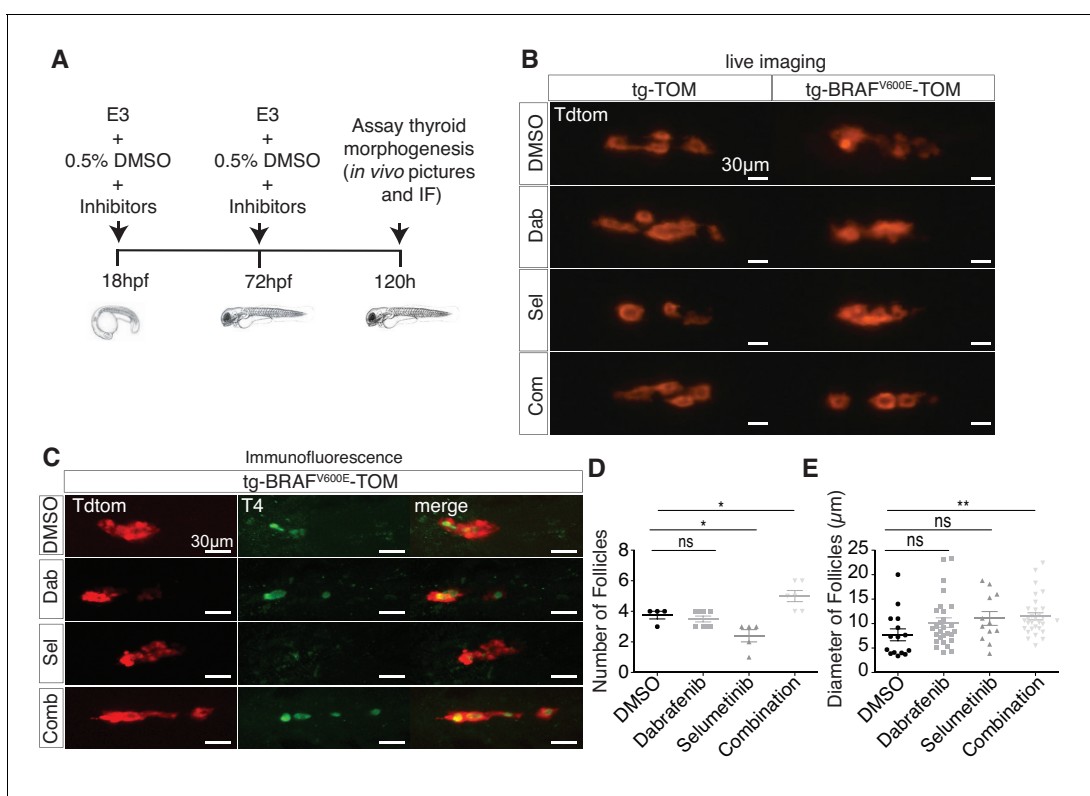

**Figure 2.** Combined treatment with BRAF and MEK inhibitors suppresses morphologic effects induced by BRAF$^{V600E}$ in thyrocytes and restores normal follicle structure. (A) Schematic diagram of experimental scheme and treatment conditions of tg-TOM and tg-BRAF$^{V600E}$-TOM larvae. (B) Epifluorescent live images of tg-TOM and tg-BRAF$^{V600E}$-TOM larvae treated with either DMSO (vehicle), 2.5 μM dabrafenib, 10 μM selumetinib, or the combination of 2.5 μM dabrafenib and 10 μM selumetinib. (C) Representative confocal images of tg-BRAF$^{V600E}$-TOM larvae treated with either DMSO (vehicle), 2.5 μM dabrafenib, 10 μM selumetinib, or the combination of 2.5 μM dabrafenib and 10 μM selumetinib and subject to immunofluorescence (IF) using anti-T4 antibody. (D–E) Quantification of thyroid follicle number (D) and follicle diameter (E) in tg-BRAF$^{V600E}$-TOM larvae after drug treatment compared to vehicle (DMSO) control. The experiment was performed in triplicate. *p≤0.05, **p≤0.01, Student's two-tailed *t*-test, ns: not significant.

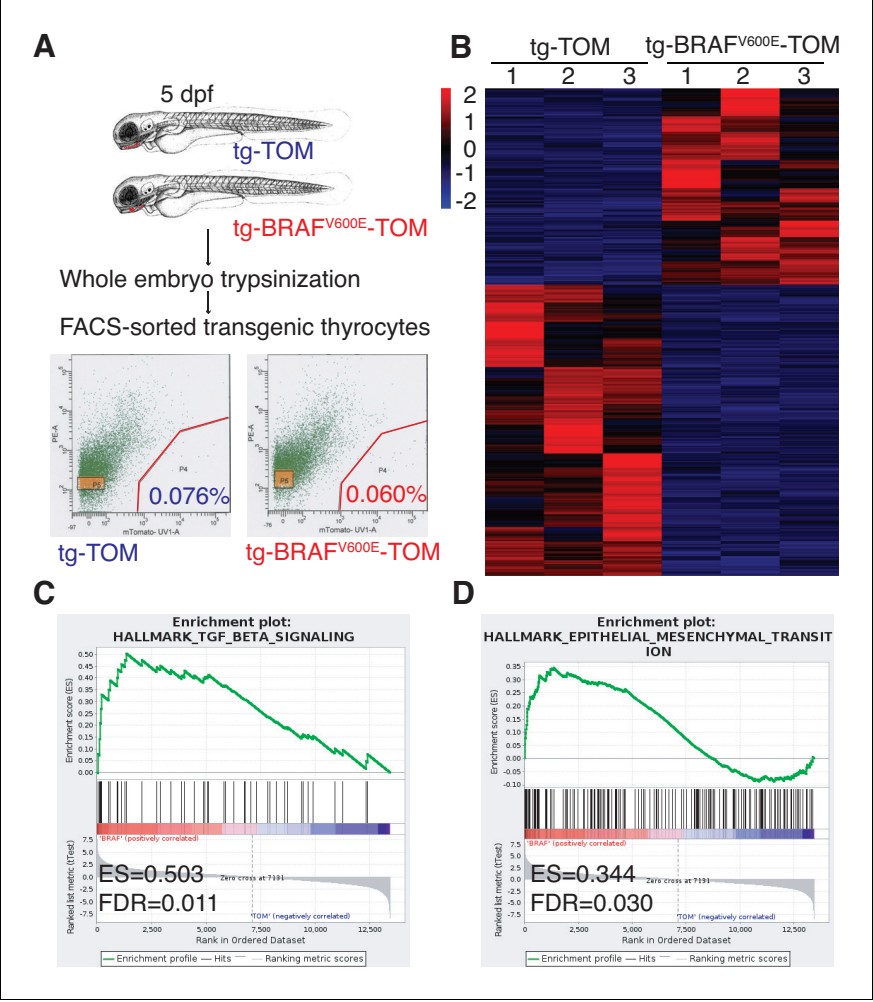

**Figure 3.** BRAF$^{V600E}$ expression in thyrocytes leads to transcriptional upregulation in genes associated with EMT and TGF-$\beta$ signaling. (**A**) Schematic diagram of the FACS-sorting workflow with the experimental plots indicating the percentage of TdTomato-positive cells sorted from tg-TOM and tg-BRAF$^{V600E}$-TOM. (**B**) Heat-map depicting top differentially expressed genes (*Figure 3—source data 1*) in sorted tg-BRAF$^{V600E}$-TOM compared to tg-TOM. (**C,D**) Gene set enrichment plots of TGF-$\beta$ and EMT signatures from RNAseq performed on sorted thyrocytes from tg-BRAF$^{V600E}$-TOM and tg-TOM larvae (E.S.=0.503, FDR = 0.011, nominal p-val = 0.002 and E.S.=0.344, FDR = 0.035, nominal p-val = 0.0, respectively).

The following source data and figure supplement are available for figure 3:

**Source data 1.** Differentially expressed genes (FDR $\leq$ 0.05, 1.2$\leq$ log$_2$FC $\leq-1.2$) in thyrocytes sorted from tg-BRAF$^{V600E}$-TOM as compared with tg-TOM in larval embryos at 5dpf.

**Figure supplement 1.** Gene expression changes induced by BRAF$^{V600E}$ in larval thyrocytes.

control and BRAF$^{V600E}$ transgenic larvae. Gene expression analysis revealed significant, differential gene expression between sorted thyrocytes from tg-TOM and tg-BRAF$^{V600E}$-TOM larvae (*Figure 3B* and *Figure 3—source data 1*). Several upregulated genes (*wnt11r, her6, axin2* and *twist3*) in tg-BRAF$^{V600E}$-TOM were validated by qPCR (*Figure 3—figure supplement 1A*). To identify key pathways affected by BRAF$^{V600E}$ expression in thyrocytes, we performed gene set enrichment analysis (GSEA) (*Subramanian et al., 2005*). This analysis revealed TGF-$\beta$ and EMT as the most highly differentially induced pathways in BRAF$^{V600E}$ expressing thyrocytes (*Figure 3C–D* and *Figure 3—figure supplement 1B*). Immunostaining of transgenic zebrafish larvae revealed a loss of E-cadherin

staining in tg-BRAF$^{V600E}$-TOM thyrocytes (*Figure 3—figure supplement 1F*). To identify similarities between zebrafish thyroid signatures and human signatures, we investigated whether there is a correlation between BRAF$^{V600E}$-induced signature and a human ERK signature (*Pratilas et al., 2009*). We found a significant positive correlation with human ERK signature genes (*Figure 3—figure supplement 1C–E*), consistent with the activation of a conserved program of gene activation with expression of oncogenic BRAF. These results demonstrate that BRAF$^{V600E}$ expression in thyrocytes induces alterations in thyroid follicle morphogenesis that are accompanied by gene expression changes in TGF-$\beta$ signaling consistent with EMT.

## BRAF$^{V600E}$ expression in thyrocytes induces thyroid cancer in adult zebrafish

In order to examine the effects of prolonged BRAF$^{V600E}$ expression, we performed a detailed histopathologic study of cohorts of transgenic tg-BRAF$^{V600E}$-TOM at 5 and 12 months post-fertilization (mpf). Five mpf tg-BRAF$^{V600E}$-TOM adult animals are characterized by thyroglobulin-positive masses invading adjacent gills and skeletal muscle, features that are not observed in age-matched wild-type tg-TOM animals (*Figure 4C–D*). Consistent with constitutive activation of BRAF$^{V600E}$, we observed a marked increase in p-ERK in thyroid tissue dissected from tg-BRAF$^{V600E}$-TOM fish compared to control thyroid tissue from tg-TOM fish (*Figure 4E*). However, at this stage, invasive thyrocytes did not harbor nuclear grooves or other cytologic features diagnostic of human papillary thyroid carcinoma. We analyzed a second cohort of tg-BRAF$^{V600E}$-TOM adult animals at 12 mpf and found evidence of progressive disease and histologic and cytological features diagnostic of thyroid carcinoma (*Figure 5*), including highly invasive carcinomas with thyrocytes invading skeletal muscle and soft tissue (*Figure 5B*). These features were not observed in control tg-TOM animals (*Figure 5B*). We also observed multiple specific cytologic features of papillary thyroid carcinoma including nuclear grooves, crowded and overlapping nuclei, and elongated nuclei (*Figure 5C*). Other features consistent with aggressive thyroid carcinoma including a desmoplastic stromal reaction (2/14 animals, 14%) were observed (*Figure 5C*). These data reveal that persistent expression of BRAF$^{V600E}$ in thyrocytes leads to invasive carcinoma with hallmarks of human PTC in 9/14 animals (64%) at one year.

## Gene expression analysis of zebrafish thyroid carcinoma reveals pathways involved in disease progression and a gene expression signature that is predictive of disease-free survival in human PTC

To identify molecular pathways involved during progression of BRAF$^{V600E}$-induced thyroid carcinoma, TdTomato-positive thyroid carcinomas were micro-dissected from the ventral jaw at 12 mpf and RNA-seq analysis was performed (*Figure 6A*). Transcriptomic analysis revealed significant, differential gene expression between normal thyroid tissue isolated from tg-TOM zebrafish and carcinomas isolated from tg-BRAF$^{V600E}$-TOM animals (*Figure 6B* and *Figure 6—source data 1*). To investigate the signaling pathways altered in zebrafish thyroid carcinomas, we performed GSEA analysis (*Figure 6C*). As observed in thyrocytes isolated from larvae, we found persistent upregulation in TGF-$\beta$ signaling, EMT, and glycolysis gene signatures from thyroid carcinomas isolated from 12 mpf zebrafish (*Figure 6C*, highlighted in red). However, in adult thyroid carcinoma we found that the top upregulated gene expression signatures are linked to proliferation and metabolism (MYC, and mTORC1 gene sets, highlighted in green). These results indicate that prolonged expression of BRAF$^{V600E}$ leads to upregulation of gene expression pathways associated with proliferation and regulation of cellular metabolism.

Consistent with some BRAF mouse models of thyroid cancer (*Charles et al., 2011*) we observed a relatively long latency of tumor development. This led us to hypothesize that thyroid carcinomas arising in zebrafish harbor gene expression changes found in indolent human papillary thyroid cancers. To examine this further, we identified a limited set of genes that are significantly deregulated in zebrafish thyroid carcinomas (*Figure 6D*, n = 58 genes highlighted in blue, *Figure 6—source data 2*) and we used this as a signature to interrogate a reference cohort of 496 human PTCs annotated with clinical outcomes data (*Cancer Genome Atlas Research Network, 2014*). We find that the gene signature identified in zebrafish stratifies disease-free survival in human patients and identifies patients at significantly lower risk of relapse (*Figure 6E*). These data imply that thyroid

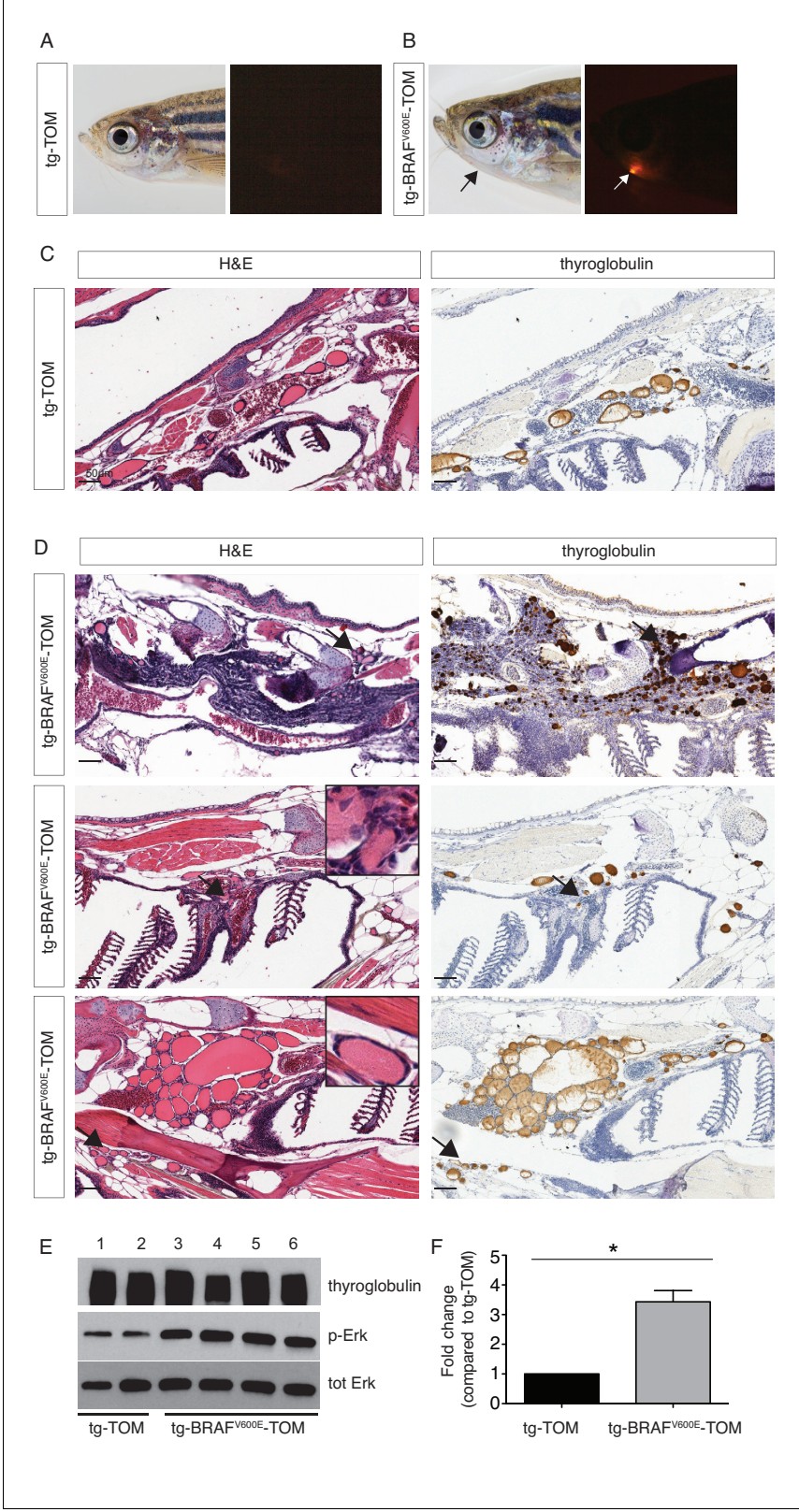

**Figure 4.** Analysis of a cohort of transgenic zebrafish expressing BRAF$^{V600E}$ in thyrocytes at 5 months post fertilization. (A–B) Photographs and epifluorescent photographs of live five mpf tg-TOM (A) and Tg-BRAF$^{V600E}$-TOM fish (B) reveals TdTomato-positive tissue. (C) Representative H&E and IHC using anti-thyroglobulin antibody of sagittal section of adult fish tg-TOM at five mpf. (D) Representative images from three tg-BRAF$^{V600E}$-TOM adult

*Figure 4 continued on next page*

*Figure 4 continued*
fish at five mpf demonstrating the presence of invasive carcinoma. Top panel, mass of Tg-positive thyroid cells (arrows) invading muscle. Middle panel, thyroid cells invading gill tissue (arrows), inset is 40X. Lower panel, thyroid cells invading subcutaneous fat in the ventral jaw (arrows), inset is 40X. (E) Western blot analysis of thyroid obtained from two tg-TOM (control) and four tumors bearing tg-BRAF^V600E-TOM zebrafish. Western blot analysis was performed in duplicate, a representative blot is shown. (F) Quantitative analysis of fold change in pERK/total ERK from (E), p<0.05, Student's t-test.

carcinomas arising in zebrafish are representative of slow growing, relatively indolent BRAF^V600E-driven PTCs arising in human patients.

## Identification of *twist3* as a critical mediator of BRAF^V600E in thyrocytes

We performed an in silico screen to identify transcription factors associated with BRAF expression in thyrocytes. Using a reference set of RNAseq data from 496 human PTCs, we performed GSEA iteratively across 1657 transcription factors to identify enrichment with genes induced by BRAF^V600E expression in larvae (*Figure 7—source data 1*). We find that the key transcriptional regulators of EMT, TWIST2 and SNAI2, are highly correlated with BRAF^V600E signature genes (*Figure 7A*, and *Figure 7—figure supplement 1*). We confirmed that *Twist3*, an orthologue of TWIST2, was significantly upregulated in BRAF^V600E expressing thyrocytes (*Figure 3—figure supplement 1*). These data indicate that Twist transcription factors are upregulated in thyrocytes after BRAF^V600E expression and may be responsible for inducing broad gene expression changes linked to disruption of thyroid follicles.

To determine the functional requirements for Twist transcription factors downstream of BRAF^V600E in thyrocytes, we performed CRISPR/Cas9 gene editing. We designed and validated an sgRNA targeting *twist3* in zebrafish embryos. Microinjection of a *twist3* targeted sgRNA in combination with recombinant Cas9 protein leads to specific indels in the *twist3* gene at the target locus, a majority of which are predicted to disrupt protein coding (*Figure 7B*). In order to examine thyroid effects, we performed CRISPR/Cas9 gene editing of *twist3* in tg-BRAF^V600E-TOM larvae. After microinjection of sgRNA targeting *twist3*, we observed rescue of follicle structure and thyroid hormone production in 57/173 (32.9%) embryos (*Figure 7C–D*). In contrast, non-targeting sgRNA-injected tg-BRAF^V600E-TOM embryos exhibit profound defects in follicle structure and thyroid hormone production (*Figure 7C*) as previously described. To determine whether the *twist3* gene was edited in thyrocytes, we flow sorted TOM+ thyrocytes from 6dpf larvae (*Figure 7E*) injected with twist3 or non-targeting sgRNAs. We isolated gDNA from TOM+ cells and performed a clonal analysis of twist3 gene editing. We identified a deletion in the twist3 gene in 1/12 clones (8.3%) from TOM+ cells isolated from embryos injected with a twist3 targeting sgRNA, and 0/12 clones isolated from TOM+ cells from embryos injected with a non-targeting sgRNA. These data indicate that genetic disruption of *twist3* is sufficient to partially rescue the effects of BRAF^V600E on thyroid follicle structure and hormone production.

## Discussion

Here, we describe the generation and characterization of a novel model of thyroid cancer. We created a transgenic zebrafish expressing BRAF^V600E in fluorescently labeled thyrocytes. By following thyrocytes from the earliest stages of specification, we are able to examine the timeline of oncogenic BRAF^V600E in vivo where we observe significant disruption of follicle structure and thyroid hormone production. However, thyrocyte number remains normal in larval development. RNAseq analysis from sorted thyrocytes confirms that genes associated with EMT and TGF-$\beta$ are transcriptionally activated by BRAF^V600E. These data suggest that BRAF^V600E activation of downstream signaling pathways converges on a gene expression program that results in loss of epithelial organization without affecting proliferation. Pharmacologic inhibition of BRAF/MEK suppresses the morphologic alterations and restores normal thyroid follicle structure. Prolonged expression of oncogenic BRAF^V600E leads to slow growing thyroid carcinoma with nuclear features similar to human PTC and gene expression changes conserved in human PTCs and predictive of improved disease free survival.

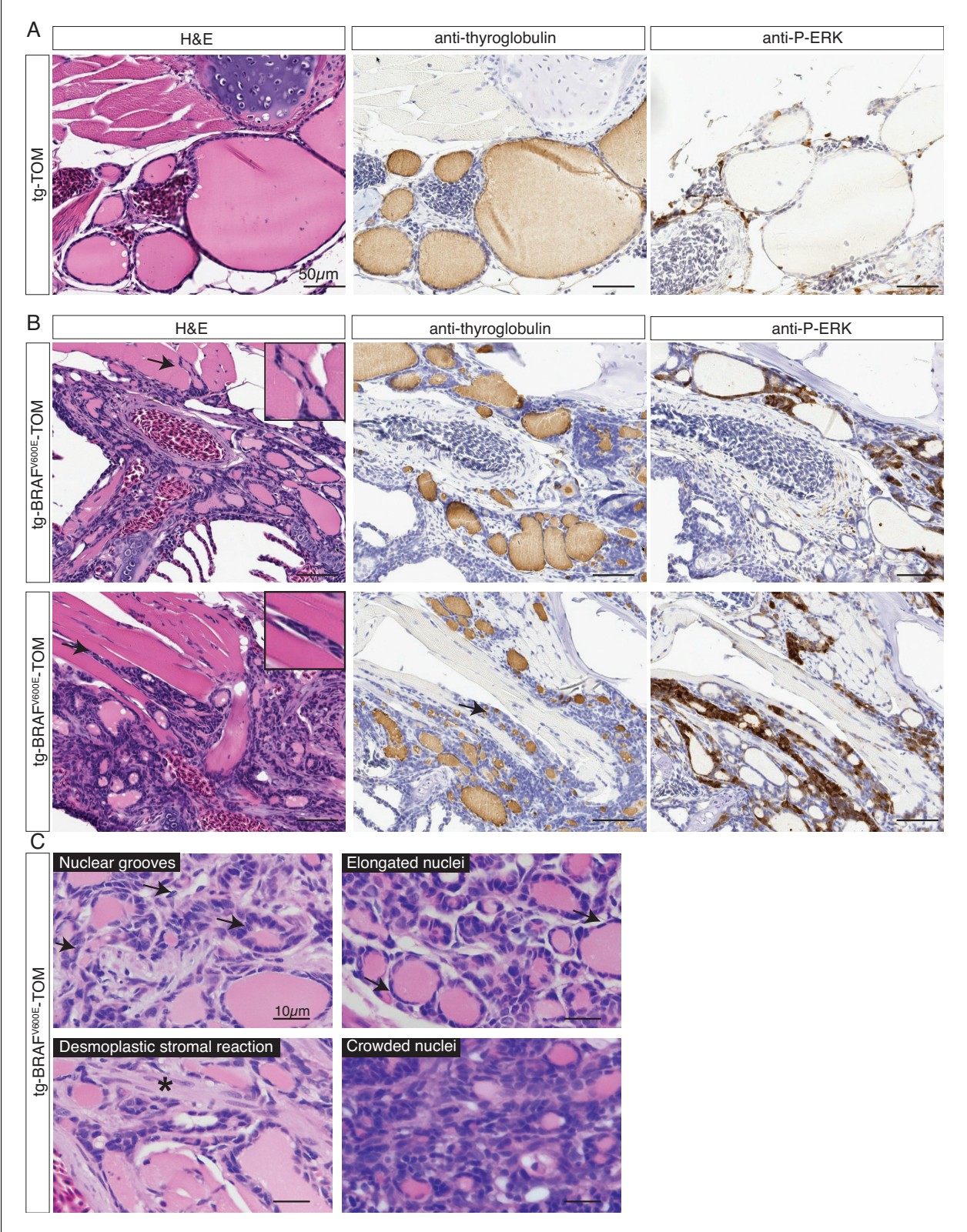

**Figure 5.** Adult transgenic fish expressing *BRAF*[V600E] in thyrocytes develop invasive thyroid cancer similar to human PTC. (A–B) Representative H&E and IHC staining using anti-thyroglobulin and anti-P-ERK antibodies performed on sagittal sections from control tg-TOM (A) and tumor bearing tg-BRAF[V600E]-TOM adult zebrafish at 12 mpf (B) (20X magnification). Arrows indicate sites of carcinoma invading skeletal muscle, inset is 40X magnification of skeletal muscle invasion. (C) High magnification photomicrographs of H&E-stained sections from tg-BRAF[V600E]-TOM animals. Key diagnostic features

*Figure 5 continued on next page*

*Figure 5 continued*

of papillary thyroid carcinoma are observed: nuclear grooves (arrows), elongated nuclei (arrows), desmoplastic stromal reaction (asterisk) and crowded nuclei (60X magnification).

We have taken advantage of the optical clarity of zebrafish larvae to examine the earliest manifestations of oncogenic BRAF$^{V600E}$ activity in vivo. Expression of BRAF$^{V600E}$ leads to a disruption of follicle structure and thyroid hormone synthesis in zebrafish. Our findings are consistent with prior

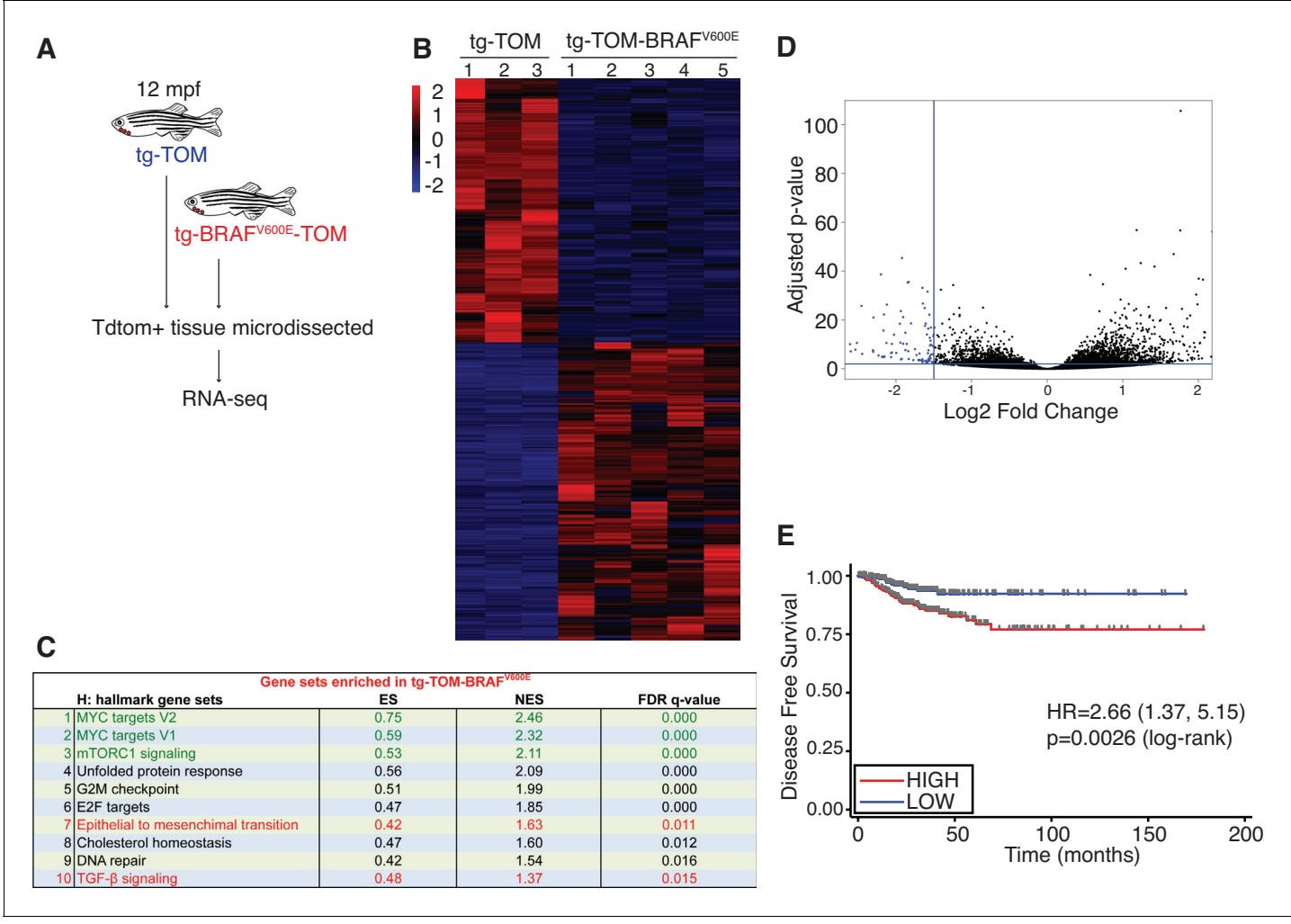

**Figure 6.** Identification of a gene expression signature from adult zebrafish thyroid carcinomas conserved in human patients with thyroid cancer. (**A**) Schematic diagram of the RNA-seq workflow from adult (12mpf) thyroid tissue. (**B**) Heat-map depicting the top differentially expressed genes in tg-BRAF$^{V600E}$-TOM thyroid tissue compared to tg-TOM thyroid tissue (*Figure 6—source data 1*). (**C**) GSEA hallmark gene sets significantly enriched in tg-BRAF$^{V600E}$-TOM. (**D**) Volcano plot depicting differentially expressed genes in tg-BRAF$^{V600E}$-TOM compared to tg-TOM thyroid tissue; a set of 58 genes (blue, FDR ≤ 1 × 10$^{-6}$, FC ≤1.5, (*Figure 6—source data 2*) was defined as a signature of BRAF$^{V600E}$ expression in thyroid carcinoma. (**E**) Kaplan-Meier analysis of disease free survival in 496 patients with papillary thyroid cancer stratified by median BRAF$^{V600E}$ gene expression signature identified in (**D**).

The following source data is available for figure 6:

**Source data 1.** Differentially expressed genes (FDR ≤ 0.05, 1.2≤ log$_2$FC ≤−1.2) in thyroid carcinomas isolated from adult tg-BRAF$^{V600E}$-TOM zebrafish as compared with thyroid tissue from adult tg-TOM zebrafish.

**Source data 2.** BRAF$^{V600E}$ downregulated genes (n = 58 genes; FDR ≤ 1×10$^{-6}$, log$_2$FC ≤1.5) identified in adult thyroid carcinomas.

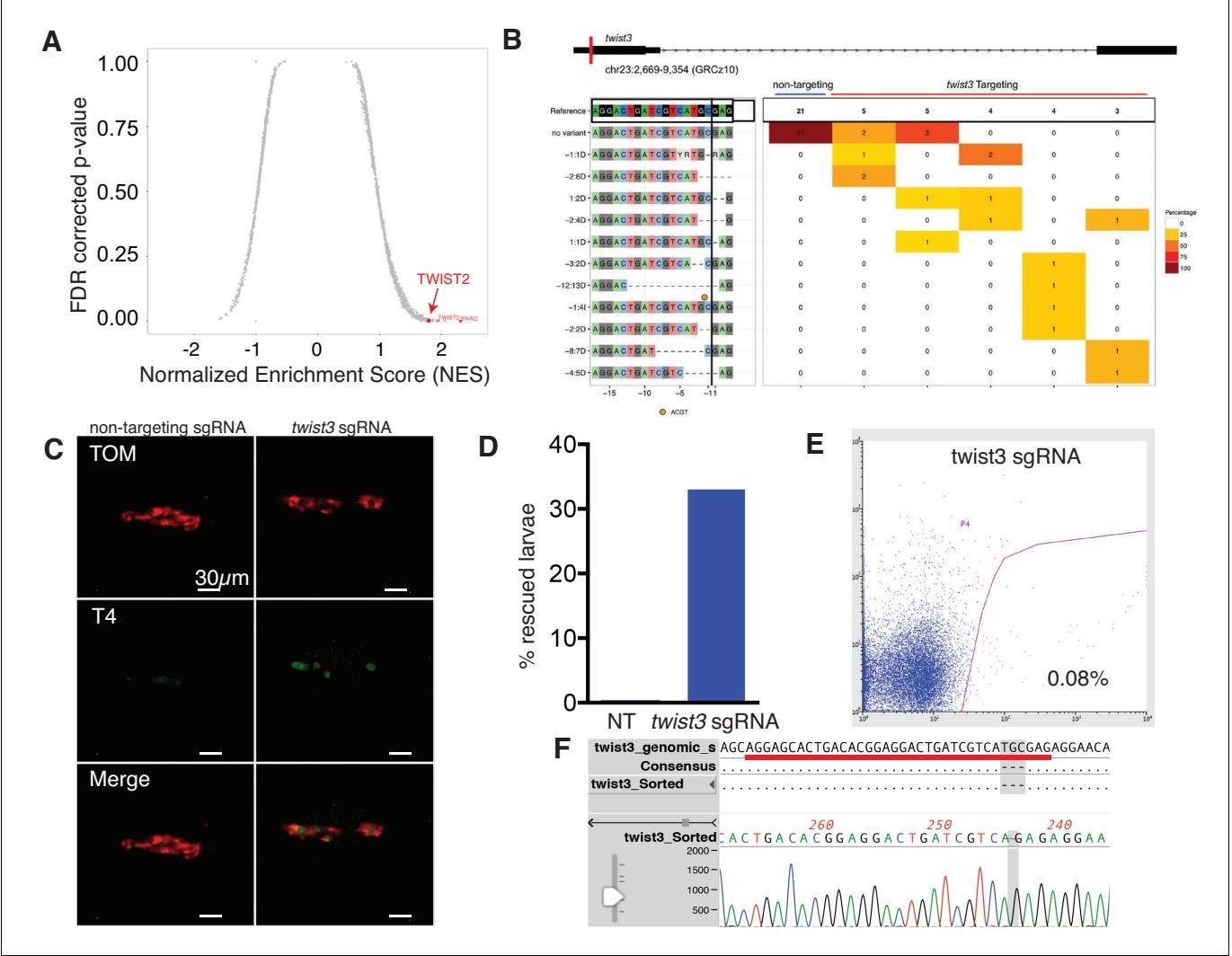

**Figure 7.** Twist expression is required for BRAF[V600E] mediated thyroid follicle disruption. (**A**) Results from an in silico screen of transcription factor (n = 1665) expression in human PTC correlated with BRAF[V600E] gene signature performed by GSEA. Top hits represented in red. (**B**) CrisprVariants analysis of *twist3* gene editing in zebrafish embryos. tg-BRAF[V600E]-TOM embryos were microinjected with rCas9 and either a non-targeting sgRNA or a *twist3*-specific sgRNA. Individual embryos were isolated and a clonal analysis of the *twist3* target locus was performed. Non-targeting embryos (n = 5, grouped) had no evidence of deletions at the *twist3* locus, 21/21 clones were wild-type. *Twist3* targeted embryos (n = 5, one per column) had a range of small indels at the targeting site. (**C**) Representative confocal images of 5 dpf fixed tg-BRAF[V600E]-TOM embryo injected with either non-targeting or *twist3* targeting sgRNA. Anti-T4 immunostaining performed to identify T4-positive (green) follicles. (**D**) Quantification of follicle rescue in embryos injected with a non-targeting versus *twist3* targeting sgRNA in BRAF[V600E]-TOM embryos. Follicle rescue was observed in 0/142 non-targeting sgRNA injected embryos vs. 57/173 (32.9%) of *twist3* sgRNA injected embryos, $p < 1.0 \times 10^{-3}$, two-tailed Fisher exact test. (**E**) FACS plot of TdTomato-positive cells isolated from *twist3* sgRNA injected tg-BRAF[V600E]-TOM embryos. (**F**) Alignment of DNA chromatogram from a clone isolated from *twist3* sgRNA injected TOM+ cells harboring a three nucleotide deletion in the *twist3* gene. The *twist3* sgRNA sequence is underlined in red.

The following source data and figure supplement are available for figure 7:

**Source data 1.** BRAF[V600E] upregulated genes (n = 85 genes; FDR ≤ 0.01, FC ≥ 1.5) identified in larval thyrocytes.

**Figure supplement 1.** Validation of gene editing in thyrocytes and identification of a BRAF[V600E] gene signature.

studies performed in cell culture using rat thyrocytes that demonstrated increased invasion and expression of markers consistent with EMT after expression of BRAF$^{V600E}$ (*Riesco-Eizaguirre et al., 2009*). In a mouse model of PTC expression of BRAF$^{V600E}$ led to the development of poorly differentiated carcinomas associated with activation of EMT (*Knauf et al., 2011*). Studies in human thyroid cancer cell lines in vitro also demonstrate that expression of BRAF$^{V600E}$ induces SNAIL, represses E-CADHERIN, and leads to increased invasion (*Baquero et al., 2013*). Gene expression studies of human papillary thyroid cancer have identified increased expression of key EMT effectors in the invasive front of tumors (*Vasko et al., 2007*). BRAF$^{V600E}$ has been found to physically interact and activate p21-activated kinase 1 (PAK1), a serine/threonine kinase involved in promoting migration in human thyroid cancer cell lines (*McCarty et al., 2014*). Our data significantly extend these studies and reveal that epithelial organization and thyroid hormone production is disrupted in vivo after expression of BRAF$^{V600E}$. In contrast to a transgenic mouse models that develops aggressive poorly-differentiated thyroid carcinomas, our zebrafish model develops relatively slow growing thyroid cancer similar to the majority of human patients with PTC that grow slowly over many years. This has allowed us to dissect early vs. late effects of BRAF signaling on thyrocytes during malignant transformation. Future studies may take advantage of this latency to test candidate genes identified in cancer genomic studies (*Landa et al., 2016*), and examine novel pathways that cooperate with BRAF$^{V600E}$ to accelerate tumor initiation and progression.

We tested the effects of small molecule kinase inhibitors, clinically used for treatment of patients with *BRAF$^{V600E}$* mutant tumors, using chemical-genetic approaches in transgenic zebrafish larvae. We find that treatment with either dabrafenib (a BRAF$^{V600E}$ inhibitor) or selumetinib (a MEK inhibitor) alone is insufficient to suppress the morphologic phenotypes induced by oncogenic BRAF$^{V600E}$ in zebrafish thyrocytes, but combined therapy with both agents suppresses oncogenic signaling and restores normal morphology. Similarly, in a *BRAF*-mutant ATC mouse model, the combination of PLX4720, a BRAF inhibitor and PD0325901, a MAPK kinase inhibitor, was more potent in suppressing MAPK activation than either agent alone (*McFadden et al., 2014*). Clinical studies with these agents in *BRAF$^{V600E}$*-mutant melanomas also confirm that combined therapy has greater anti-tumor efficacy in human patients (*Long et al., 2014*). Multiple pathways that mediate resistance to BRAF inhibitors have been characterized in melanoma and colorectal carcinoma (*Pritchard and Hayward, 2013*). Our findings lead us to speculate that combined therapy with BRAF and MEK inhibitors may be required for treatment of *BRAF$^{V600E}$* mutant thyroid carcinoma. This hypothesis is being tested in an ongoing clinical trial examining BRAF inhibition alone versus the combination of BRAF inhibition and MEK inhibition (ClinicalTrials.gov identifier NCT01723202).

Prolonged expression of BRAF$^{V600E}$ leads to an invasive thyroid carcinoma with diagnostic features of human papillary thyroid carcinoma in adult zebrafish. Gene expression studies from tumors isolated from adult animals reveal signatures of proliferation and alterations in mTOR signaling during progression. In a mouse model of thyroid cancer genetic inactivation of TSH receptor attenuated tumor initiation suggesting that TSHR signaling cooperates with oncogenic BRAF (*Franco et al., 2011*). We suggest that zebrafish thyroid carcinomas may arise slowly due to a relative euthyroid state that does not provoke elevated TSH levels, as seen in mouse models. Alternatively, it is possible that additional genetic alterations may be required for development of carcinoma. Thyroid cancers in zebrafish exhibit similar histologic features to human papillary thyroid cancer. These are relatively slow growing carcinomas that invade local tissues and exhibit a desmoplastic stromal reaction. Key diagnostic features of PTC, such as nuclear grooves and elongation, are observed, indicating conservation of these phenotypes across broad evolutionary distance. Interestingly tumor formation in zebrafish thyroid occurs on a tp53 wild-type background, in contrast to a zebrafish melanoma model in which tp53 loss of function is required for tumor formation (*Patton et al., 2005*). These results confirm the importance of cellular context in modulating tumor initiation, consistent with genetic studies from human patients, which demonstrate BRAF mutations in papillary microcarcinomas (*Li et al., 2015*), the earliest pathologic lesions identified.

By studying the gene expression changes induced by BRAF$^{V600E}$, we found enrichment of TGF-$\beta$/EMT signatures and a correlation with TWIST2 expression in human papillary thyroid carcinomas. TWIST proteins are over-expressed in a significant fraction of thyroid cancers (*Wang et al., 2013*) and knockdown of TWIST1 in thyroid cancer cell lines led to a reduction in proliferation and migration (*Salerno et al., 2011*). We find that *twist3*, an orthologue of *TWIST2*, is significantly upregulated in thyrocytes upon *BRAF$^{V600E}$* expression. Using CRISPR/Cas9 gene editing we find that genetic

inactivation of *twist3* suppressed the effects of oncogenic BRAF[V600E] in thyrocyte follicle structure and hormone synthesis. To our knowledge, these are the first in vivo data demonstrating a genetic requirement for TWIST expression downstream of BRAF[V600E] in a thyroid cancer model.

## Materials and methods

### Generation of constructs

All vectors were created using Gateway recombination (Life Technologies, Carlsbad, CA). A human *BRAF*[V600E] middle entry clone was made by PCR amplification as previously described (*Ceol et al., 2011*). pME-*BRAF*[V600E], and pME-*TdTomato*, were used in LR reactions with pDestTol2pA, p3E-polyA and p5E-*tg* (containing a 514 bp of zebrafish thyroglobulin promoter fragment, as described (*McMenaminetal., 2014*) to generate pTol2*tg:TdTomato-polyA,* pTol2*tg:EGFP-polyA*, and pTol2*tg: hBRAFV600E-polyA.*

### Generation of zebrafish transgenic line

Twenty-five picograms of pTol2tg:TdTomato-pA and 25 pg mRNA encoding the Tol2 transposase were injected into one-cell stage wild type AB larvae to create Tg(*tg:TdTomato-pA)* transgenic line. F1 TdTomato-positive animals were raised to adulthood and crossed to wild type fish. To create Tg (*tg:BRAF*[V600E]*-pA;tg:TdTomato-pA*) line, transgenic founders (F$_0$) were generated by co-injecting 25 pg pTol2tg:TdTomato-pA with 25 pg pToltg:BRAF[V600E]-pA and 25 pg mRNA encoding the Tol2 transposase. Injected larvae were raised to adulthood and crossed to wild type fish for transgenic screening. F$_1$ animals were raised to adulthood and genotyped for the presence of *BRAF* via PCR. Identification of transgenic Tg(*tg:BRAF*[V600E]*-pA;tg:TdTomato-pA*) line was performed using PCR-based amplification of *BRAF* in F2 TdTomato-positive larvae. To simplify nomenclature, we refer to larvae/adult Tg(*tg:TdTomato-pA*), Tg(*tg:BRAF*[V600E]*-pA;tg:TdTomato-pA*) as tg-TOM, and tg-BRAF[V600E]-TOM, respectively.

### Genotyping

Adult zebrafish were anesthetized with 0.004% MS-222 (Tricaine) (Western Chemical, Ferndale, WA), fin clipped and tissue was incubated in 150 µl of PCR lysis solution containing 20 µg/ml Proteinase K (Qiagen, Hilden, Germany) at 55°C for 60 min and followed by 45 min at 85°C. At the end of the incubation, lysates were centrifuged at 3000 rpm for 5 min and 100 µl were transferred to a new tube. One µl was used in a 10 µl PCR reaction using primers listed in *Supplementary file 1*.

### Cell counting

Five-day-old larvae were sacrificed and fixed in 4% PFA ON at 4°C washed in PBS containing 0.1% Tween 20 (PBST), treated with 50 µg/ml proteinase K (Roche, Basel, Switzerland) for 35 min and incubated with Sytox Green Nucleic Acid stain (1:1000, Thermo Fisher Scientific, Waltham, MA) for 1 hr at RT. Larvae were washed with PBST and post-fixed in 4% PFA for 20 min. After several washes with PBST, larvae were mounted in agarose. Z stacks were acquired on a confocal microscope and green-stained nuclei of TdTomato-positive cells were counted in three-dimensional (3D) image stacks using Imaris software (Bitplane, Concord, MA).

### BrdU incorporation

Three-day-old embryos were incubated in E3 fish water containing 10 mM BrdU (Sigma, St Louis, MO) and 15% DMSO for 20 min at 4°C. Medium was replaced with fresh E3 and embryos were incubated for 2 days at 28°C. Larvae were fixed in 4% PFA for 2 hr at RT, treated with 50 µg/ml proteinase K (Roche, Basel, Switzerland) for 35 min, post-fixed in 4% PFA for 20 min and incubated in 2N HCl for 1 hr. HCl solution was then removed and larvae were incubated in 0.1M Borate buffer, pH 8.5 for 20 min at RT. After several washes in PBT, larvae were incubated with blocking buffer and anti-BrdU antibody (1:200, Sigma, St Louis, MO) as described in Immunofluorescence.

### Inhibitor treatment

Larvae were incubated at 18-somite stage in a 1 mL E3 medium containing 0.5% DMSO in a 24-well plate in the presence of the following inhibitors: 2.5 µM Dabrafenib (Selleckchem, Houston, TX), 10

μM Selumetinib (Selleckchem, Houston, TX) or a combination of the two inhibitors. These were the maximal tolerated doses compatible with normal development.

## Imaging

Micrographs of whole larvae and adult fish were acquired with a Zeiss Discovery V8 stereomicroscope (Zeiss, Oberkochen, Germany) equipped with epifluorescence and appropriate filters. Live imaging of fluorescence larvae was acquired with a Zeiss LS510 laser scanning confocal microscope (Zeiss, Oberkochen, Germany). Images were further analyzed for detailed quantitative 3D analysis using Imaris software (BitPlane, Concord, MA). Brightfield images of adult fish were captured using a Canon EOS60D camera (Canon, Tokyo, Japan).

## Sample preparation and histopathology

Five (n = 5 and n = 10 tg-TOM and tg-BRAF$^{V600E}$-TOM, respectively) or 12-month old (n = 5 and n = 14 tg-TOM and tg-BRAF$^{V600E}$-TOM, respectively) animals were sacrificed and fixed in 4% PFA at 4°C for 3 days, followed by 3 days of decalcification in EDTA 0.5 M pH 8. After dehydration in 70% ethanol and processing using an automatic tissue processor, samples were embedded in paraffin and 5 μm sections were mounted on slides and stained with H&E for histopathologic examination. H&E-stained slides were reviewed with a board certified human pathologist specializing in the diagnosis of thyroid cancer (TS). The specific histopathologic features were reviewed and validated in comparison with normal thyroid specimens.

## RNA extraction

Twelve-month-old animals were sacrificed, and TdTomato-positive thyroid tissue was dissected under a stereomicroscope and placed in 500 μL Trizol (Life Technology, Carlsbad, CA). After homogenization, 100 μL of chloroform was added, and the samples were centrifuged at 12,000 x g for 15 min at 4°C. After centrifugation, the aqueous phase was transferred to a new tube, 300 μl of 70% ethanol was added and purified using an RNeasy Mini spin column (Qiagen, Hilden, Germany) as described by the manufacturer, including DNAse treatment (Qiagen, Hilden, Germany).

## Western blot

Five-month-old animals were sacrificed and TdTomato-positive thyroid tissue was dissected under a stereomicroscope and homogenized in 40 μl RIPA buffer (Pierce, Waltham, MA). Five μg of total extract was resolved by SDS-PAGE, transferred to nitrocellulose and probed with the following antibodies: Phospho-p44/42 (1:1000, Cell Signaling, Danvers, MA), p44/42 (1:1000, Cell Signaling, Danvers, MA), thyroglobulin (1:20000, Dako, Glostrup, Denmark).

## Immunohistochemical analysis

Sections were deparaffinized with xylene and washed with ethanol followed by antigen retrieval using the following conditions: H$_2$O$_2$ at pH 9 at 100°C for 20 min. All immunostains were performed on the Bond III Autostainer (Leica Biosystem, Wetzlar, Germany) using the Bond Polymer Refine Detection kit (Leica Biosystem, Wetzlar, Germany) using the following conditions: blocking for 10 min, anti-TG antibody 1:2000 (Dako, Glostrup, Denmark), anti-PCNA 1:1000 (Sigma, St. Louis, MO) or anti-P-ERK 1:1000 (Cell Signaling, Danvers, MA) for 15 min. Slides were incubated in post primary AP for 8 min, followed by incubation for 8 min in Polymer AP. Slides were incubated with DAB for 10 min, counter stained with hematoxylin for 5 min and scanned using Aperio slide scanner (Leica, Wetzlar, Germany).

## Immunofluorescence

For anti-T4 immunofluorescence, 5-day-old larvae were fixed in 4% PFA overnight at 4°C. They were then washed in PBS containing 0.1% Tween 20 (PBST), treated with 50 μg/ml proteinase K (Roche) for 35 min and post-fixed in 4% PFA for 20 min. After several washes in PBST, larvae were incubated in blocking buffer (PBST containing 1.0% DMSO, 5% goat serum, and 0.8% Triton X-100) for 2 hr at RT, followed by overnight incubation in with anti-T4 antibody (1:2000, MP Biochemical, Solon, OH). After three washes in PBST containing 1% BSA, larvae were incubated with Alexa Fluor 488-conjugated anti-rabbit IgG secondary antibody for 3 hr at RT (1:500; Invitrogen, Carlsbad, CA). Stained

larvae were washed in PBST and embedded in 1% low melt agarose for confocal imaging. For anti-E-Cadherin staining 5-day-old larvae were frozen and embedded in OCT (Sakura Finetek, Torrance, CA). Larvae were sectioned at 10 µM on a cryotome and sections were stained with anti-E-cadherin (1:500, Abcam, Cambridge, MA), followed by anti-mouse Alexa Fluor 488.

## Embryo dissociation and fluorescence activated cell sorting

Sixty 5 dpf larvae (in triplicate for each transgenic line, tg-TOM and tg-BRAF$^{V600E}$-TOM) were sacrificed, de-yolked using calcium-free ringer solution, transferred to a 60 ml culture dish containing 4 ml of protease solution (0.25% trypsin, 1 mM EDTA) and incubated at 28°C for 30 min. The reaction was stopped with 100 µl 100 mM CaCl$_2$ and 500 µl FBS. Cells were centrifuged for 3 min at 3000 rpm and resuspended in 5 ml of suspension medium (Colorless Leibovitz L-15, 0.8 mM CaCl$_2$, penicillin/streptomycin and 1% FBS), centrifuged for 3 min at 3000 rpm, resuspended in 5 ml suspension medium and homogenized with a 10 ml pipette. Twenty U/ml of RNAse inhibitor (Life Technologies, Carlsbad, CA) was added, and the cell suspension passed through a cell strainer tube (BD Bioscience, San Jose, CA). TdTomato-positive cells were sorted using BD FACS Vantage SE with DiVa upgrade cell sorter (Becton, Dickinson and Co., San Jose, CA). For excitation of TdTomato fluorescent protein, 514 nm wavelength of Ar-Kr-ion laser with 200 mW power was used and fluorescence was detected in logarithmic mode via a BP575/26 nm optical filter. TdTomato-positive and negative cells were collected in 250 µl RLT buffer (Qiagen, Hilden, Germany) and nucleic acids were extracted. RNA extraction was performed using an RNeasy Micro Kit (Qiagen, Hilden, Germany) and subjected to DNAse treatment (Qiagen, Hilden, Germany). DNA extraction for clonal analysis of CRISPR/Cas9 activity was performed with an AllPrep DNA/RNA Kit (Qiagen, Hilden, Germany). The quality and the concentration of nucleic acids were assessed using a Bioanalyzer (Agilent, Santa Clara, CA). The RNA samples used in the study had a RIN between 7.4 and 9.4.

## Amplification-library preparation and RNA-seq

One ng of RNA from three biological replicates of sorted cells for each transgenic line tg-TOM and tg-BRAF$^{V600E}$-TOM was used to prepare amplified ds cDNA using Ovation RNA-Seq System V2 (Nugen, San Carlos, CA). Amplified ds cDNA was purified using QIAquick PCR purification kit (Qiagen, Hilden, Germany) and 200 ng of amplified cDNA was fragmented in a final volume of 50 µl using S220 Focused-ultrasonicator (Covaris, Woburn, MA), to obtain 150 bp DNA fragment size (peak incident power: 175W, duty factor: 10%, cycles per burst: 200, time: 280s). Fragmented DNA samples were used to prepare the library using TruSeq RNA sample preparation kit v2 (Illumina, San Diego, CA). For qPCR validation cDNA was diluted 1:7 and 2 µl were used in a 10 µl reaction using the PerfeCTa SYBR Green FastMix (Quanta Biosciences, Gaithersburg, MD). *Gapdh* was used as a control gene for normalization. Relative gene expression among samples was determined using the delta$C_t$ method ($2 – \Delta\Delta C_t$). Results are expressed as the mean ± SEM in relative expression. For the library preparation of RNA derived from adult animals, RNA was extracted from dissected thyroid tissues as described and 100 ng from three tg-TOM and five tg-BRAF$^{V600E}$-TOM biological replicates was used to prepare the library using TruSeq RNA sample preparation kit v2 (Invitrogen, San Diego CA). After validation using an Agilent DNA 1000 LabChip (Agilent, Santa Clara, CA), samples were submitted for RNA-seq.

## CRISPR/Cas9 experiments

Gene editing of *twist3* was performed by microinjection of 500 pg recombinant S. pyogenes Cas9 (PNA Bio, Thousand Oaks, CA) complexed with 1 µg synthetic sgRNA (ALT-R, Integrated DNA Technologies, Coralville IA). Ribonucleotide-protein complex formation was performed in vitro as described (*Zuris et al., 2015*). DNA was extracted from single injected embryos at 24hpf for analysis of indels. The targeted locus in *twist3* was PCR amplified using primers listed in *Supplementary file 1* and subject to TA cloning in pcr2.1 (Invitrogen). Plasmid DNA was isolated from individual clones and Sanger sequencing was performed. CRISPR/Cas9 gene editing was analyzed using CrispRVariants (*Lindsay, 2016*). A T7 Endonuclease I (T7EI) assay was performed using 8 µl of PCR product in 20 µl final reaction containing 2U of T7EI (NEB, Ipswich, MA) for 1 hr at 37°C. Products were analyzed on a 1% agarose gel.

## Bioinformatic analyses

Paired end reads were aligned using Star v2.3 (*Dobin et al., 2013*) to the zebrafish genome (GRCz10) using the Ensembl transcriptome (*Howe et al., 2013*). Analysis of differential gene expression was performed using DESeq2 (*Love et al., 2014*). To identify differentially regulated genes between samples, we selected genes with log2 fold change greater than 1.2 or less than −1.2 and an adjusted p-value<0.05. Orthology to human genes was determined using Ensembl (*Collins et al., 2012*) and supplemented by performing BLAST (*Altschul et al., 1990*). GSEA (*Subramanian et al., 2005*) was performed on RNAseq data using normalized counts and queried against the Hallmark in Cancer signatures from the MSigDB (http://www.broadinstitute.org/gsea/msigdb/index.jsp). Signatures represented in the results are the top 10, FDR < 25%, nominal p-value<5%. For the analysis of disease-free survival in human patients with PTC, we used clinical data from the TCGA PTC cohort (*Cancer Genome Atlas Research Network, 2014*). We utilized the adult BRAF$^{V600E}$ signature (n = 58 genes, *Figure 6D* and *Figure 6—source data 2*) to rank genes in the TCGA PTC samples using normalized RNAseq gene expression with a minimum rank method to resolve ties. We calculated the mean rank for the set of signature genes for each subject to estimate the overall regulation of the signature with respect to overall gene expression. We stratified subjects by median rank of the signature and performed a Kaplan-Meier survival analysis using disease free survival as the primary outcome. For the in silico screen to identify transcription factors associated with BRAF$^{V600E}$ gene expression signatures, we utilized normalized gene expression from 496 TCGA PTC samples described (*Cancer Genome Atlas Research Network, 2014*). We performed iterative GSEA against all transcription factors to identify enrichment with a BRAF$^{V600E}$ signature identified in larval thyrocytes (*Figure 7—source data 1*).

## Zebrafish husbandry

Zebrafish were bred and reared according to established guidelines (*Westerfield, 2007*). All studies were performed in compliance with a protocol approved by our Institutional Animal Care and Use Committee (IACUC).

## Acknowledgements

We are grateful to members of the Houvras laboratory for critical discussion and manuscript review. We gratefully acknowledge Sarah K McMenamin and David M Parichy for providing the zebrafish thyroglobulin promoter construct. We thank Todd Evans and Marco Seandel for critical reading of the manuscript. We thank Jose Cardon Costa and aquatics staff for expert zebrafish husbandry. We thank the Weill Cornell Medicine Optical Microscope Core staff for assistance with confocal imaging. We thank the Weill Cornell Medicine Electron Microscopy & Histology Core facility staff for assistance in preparing histology samples. We thank Chantal Pauli and Theresa McDonald for assistance with zebrafish histology. We thank the Flow Cytometry and Confocal Microscopy Facility at the Hospital of Special Surgery staff for their assistance with cell sorting. Next generation sequencing was performed in the Weill Cornell Medicine Epigenomics Core Facility. We thank Yanwen Jiang and Caitlin Bourque for technical help in performing RNA-seq. This work was supported by the Department of Surgery, Weill Cornell Medical College (YH); a SPORE in Thyroid Cancer, National Institute of Health (NIH) (P50-CA172012) (JV); the Medical Scientist Training Program of General Medical Sciences of the NIH (T32GM007739) to the Weill Cornell/Rockefeller/Sloan-Kettering Tri-Institutional MD-PhD Program (RM). Z Chen was partially supported by the Clinical and Translational Science Center at Weill Cornell Medical College (UL1-TR000457-06).

## Additional information

### Funding

| Funder | Grant reference number | Author |
| --- | --- | --- |
| National Institutes of Health | R21 CA202540 01 | Yariv Houvras |
| National Institutes of Health | T32GM007739 | Raul Martinez-McFaline |
| National Institutes of Health | P50-CA172012 | Jacques A Villefranc |

The funders had no role in study design, data collection and interpretation, or the decision to submit the work for publication.

## Author contributions

VA, Conceptualization, Formal analysis, Investigation, Methodology, Writing—original draft, Writing—review and editing; JAV, ER, Investigation, Methodology, Writing—review and editing; SC, Resources, Data curation, Formal analysis, Writing—review and editing; RM-M, Data curation, Investigation, Methodology, Writing—review and editing; AN, Investigation, Writing—review and editing; AV, Data curation, Software, Methodology, Writing—review and editing; RB, Data curation, Methodology; ZC, Formal analysis, Supervision, Writing—review and editing; TS, Formal analysis, Writing—review and editing; OE, Software, Formal analysis, Investigation, Writing—review and editing; YH, Conceptualization, Data curation, Formal analysis, Supervision, Investigation, Writing—original draft, Writing—review and editing

## Author ORCIDs

Yariv Houvras, http://orcid.org/0000-0003-0751-3215

## Ethics

Animal experimentation: This study was performed in strict accordance with the recommendations in the Guide for the Care and Use of Laboratory Animals of the National Institutes of Health. All of the animals were handled according to approved institutional animal care and use committee (IACUC) protocols (#2011-0026) of Weill Cornell Medical College.

# Additional files

## Supplementary files

• Supplementary file 1. Primers used for qPCR, genotyping, and gene editing.

## Major datasets

The following previously published dataset was used:

| Author(s) | Year | Dataset title | Dataset URL | Database, license, and accessibility information |
|---|---|---|---|---|
| Houvras Y, Anelli V | 2017 | Oncogenic BRAF disrupts thyroid morphogenesis and function via Twist expression | https://www.ncbi.nlm.nih.gov/geo/query/acc.cgi?acc=GSE97096 | Publicly available at the NCBI Gene Expression Omnibus (accession no: GSE97096) |

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
