## [Decision Letter]

Thank you for submitting your article "Oncogenic BRAF Disrupts Thyroid Morphogenesis and Function via Twist Expression" for consideration by *eLife*. Your article has been reviewed by three peer reviewers, and the evaluation has been overseen by Michael R. Green as the Reviewing Editor and Kevin Struhl as the Senior Editor. The following individuals involved in review of your submission have agreed to reveal their identity: Leonard I Zon, Craig Ceol.

The reviewers have discussed the reviews with one another and the Reviewing Editor has drafted this decision to help you prepare a revised submission.

Summary:

Papillary Thyroid Cancers (PTC) can present as aggressive tumors (characterized by poorly differentiated cells) or indolent tumors (well-differentiated cells). Existing mouse models of PTCs depict the aggressive version of the disease. In this manuscript, the authors have developed a zebrafish model for PTC that closely mimics slow growing human PTCs. Zebrafish with thryocyte-specific expression of BRAFV600E have disrupted thyroid follicle structure and hormone synthesis during development. These transgenic zebrafish, upon adulthood, develop PTCs. The timing of tumor onset, histopathology and gene expression signature of these tumors recapitulate that of slow-growing human PTCs. The authors have demonstrated that the expression profile of these tumors is predictive of disease-free survival in patients with PTCs. CRISPR/Cas9-mediated targeting of twist3 (ortholog of human TWIST2) in BRAFV600E-expressing zebrafish thyrocytes restored thyroid structure and hormone producing ability. The authors have identified twist3 as a key downstream effector of BRAFV600E induced EMT in thyrocytes and claim that this process is essential in the early oncogenic transformation that initiates these cancers.

The manuscript describes a new zebrafish model of PTC that may be the best existing representation of slow-growing human PTC. The zebrafish model of PTC may serve a useful tool for stratifying PTC patients based on disease recurrence. Experiments are performed in a logical order and the manuscript is nicely written.

Essential revisions:

1) Given that low degree of detected mutations (19%) and of these only 2 show loss of function (2/46 = 4%) in the cohort sequenced, it is surprising that the twist3 sgRNA causes such a dramatic reversal of phenotype. It is possible that the sequencing method used is underestimating the number of loss of function alleles. Please assess twist protein levels either through western or immunostaining at any time point to confirm significant loss of function of twist3. Alternatively, it the thyroid cells could be sorted at day 5 dpf after imaging and these cells could be sequenced via Next Generation Sequencing to assay the mutation rate. In addition, a negative control gRNA should be injected to ensure that injection of gRNA alone is not affecting the phenotype, especially given the discordance between the alleles predicted to be loss of function and the strong suppression of BRAF-related thyroid dysmorphic changes.

2) The authors have used tissue-specific knockout of twist3 to claim that it is necessary for BRAF^V600E^-induced morphological changes in thyroid structure. Embryos injected with twist3 sgRNA (Figure 7—figure supplement 1) have morphological defects, notably cyclopia, that are reminiscent of general RNA toxicity. The authors should show that the rescue of thyroid structure is not due to non-specific morphological changes caused by RNA toxicity. A variety of control experiments can be performed to exclude this possibility.

3) The Abstract asserts that the data demonstrate a genetic requirement for TWIST2 in the earliest step of BRAF^V600E^ transformation. Please rewrite this sentence with a more limited and specific conclusion. Alternatively, if data is already collected regarding whether twist3 CRISPR suppressed onset of BRAF-driven adult thyroid cancer in this model and if existing human sequencing data suggest this could be an early event in PTC development, please add these two lines of data.

4) There are instances where the authors should fortify the claims they are making. First, in the Abstract it is stated that there is a 'genetic requirement for TWIST2 in the earliest step of BRAF^V600E^-mediated transformation'. While twist3 might be important for BRAF^V600E^-driven changes in thyroid morphology, it is not clear that these morphological changes represent an early stage or step of oncogenic transformation. It would bolster their case if the authors show an effect of twist3 loss on BRAF^V600E^-driven tumorigenesis. Can they show that animals with thyroid-specific loss of twist3 also have delayed tumor onset? If not, then the authors' claims should be rephrased. Second, the sentence 'We identify an orthologue of human TWIST2, twist3, as a key mediator of BRAF^V600E^ induced EMT in thyrocytes' has similar issues. While an EMT signature (among others) is seen in BRAF^V600E^-expressing thyrocytes, the authors should do more before unequivocally stating this is EMT. Showing changes in expression of canonical EMT markers (e.g. E cadherin, N cadherin) is necessary.

5) All three reviewers also agree that the authors have not done the experiments to be able to conclude that there is a "pivotal functional role for TWIST expression during the earliest stage of malignant transformation by oncogenic BRAF". The conclusions should be softened in the text. The work would form the basis of a nice new paper. To support this conclusion the authors would need to do a variety of human sequencing and cell culture work. We feel that all these experiments are clearly beyond the two-month time limit recommended by *eLife*. For example, sequencing of human PTCs for early events would be a whole other collaboration/project and cell culture work may not recapitulate the indolent nature of human PTC modeled in this paper and could take a long time, may up to a year. Given the time restraints suggested by *eLife*, we are comfortable instead having the authors remove the sentence at the end of the Discussion.

Non-essential revisions:

1) One deficiency of the current manuscript is lack of human disease equivalent for the presented results. If available, please add data using human samples of thyroid cancer or thyroid cancer cell lines that this pathway is important, conserved and relevant to human thyroid cancer driven by BRAF^V600E^.

2) The authors show that the effect of BRAF^V600E^ in thyroid morphogenesis can be rescued by combined treatment with BRAF and MEK inhibitors but not by either of them individually. The authors do not explain any further why this happens. Have the authors tried higher concentrations with individual inhibitors to see if they can show rescue?

3) This is not necessarily a 'concern', but could the authors determine whether their zebrafish PTC signature aligns more closely with human (or mouse) PTC versus PDTC. If the zebrafish signature is more similar to PTCs, then it would help their argument that they are modeling a more indolent form of the disease.

4) The western blots in Figure 4 should be quantified as the changes are subtle.

5) The histology images in Figure 5 are difficult to see. Zoomed in insets of certain important regions/features within the PTC will help in visualizing these.

6) The chromatograms in Figure 7 and Figure 7—figure supplement 1 show wide peaks in mutated regions of the twist3 gene. In the clonal analyses described, where only one copy of the gene is being sequenced, regular-width peaks along with a commensurate frameshift downstream of the indel would be expected. It is not clear why the chromatograms look like they do.

7) Please describe in the text how was the gene signature used in the data presented in Figure 6 chosen. Did this gene signature predict improved OS for patients with PTC? If not, please hypothesize regarding any discrepancy.

8) Please comment in the text or Discussion why in Figure 2 selemetinib decreases number of follicles significantly.

9) Figure 4 and Figure 5 and the text associated with them would be clarified by addressing the following items.

a) What is the significance if the pathologic features shown in Figure 5 in human PTC if any?

b) Please describe in the text whether the one-year animals get metastases.

c) Were the animals with the desmoplastic reaction the same animals with the other pathologic features?

d) If the animals look different at 12m with the tumors as compared to 5 mpf, can a photo of 12mpf fish with thyroid cancer be included at the beginning of Figure 5?

e) Please also include more up close photos of the cytologic features in Figure 5, especially of the nuclear grooves and elongated nuclei so these can be appreciated.

f) Please also annotate the structures being invaded in histologic structures in Figure 4 and Figure 5 to ensure non-zebrafish audience can appreciate the data.

10) The manuscript would benefit from several small edits in order to clarify the work. Most of these edits will help the work be more accessible to a non-zebrafish audience.

Text edits:

Abstract: Please clarify that developmental effects of BRAF^V600E^-driven changes in thyroids were reversed with drug treatment.

Introduction, second paragraph: Please define the acronym NIS.

Subsection “BRAF^V600E^ expression leads to TGF-β and EMT gene expression changes in thyrocytes”: Please name the tumors that the cell lines were derived from (thyroid?) and clarify how an ERK signature is derived by inhibiting MEK. Alternatively, simplify the sentence.

Subsection “BRAF^V600E^ expression leads to TGF-β and EMT gene expression changes in thyrocytes”: Please clarify which of these groups of upregulated genes were also seen in the human data.

Subsection “BRAF^V600E^ expression in thyrocytes induces thyroid cancer in adult zebrafish”: “We analyzed a second cohort of tg-BRAF^V600E^-TOM adult animals at 12 mpf and found evidence of progressive disease and histologic and cytological features consistent with thyroid carcinoma (Figure 5).”: This sentence is redundant with the following sentence, please remove or combine with the following sentence. Or clarify the point being made.

Subsection “BRAF^V600E^ expression in thyrocytes induces thyroid cancer in adult zebrafish”: The last sentence is also redundant, please remove this sentence or clarify if a different point was meant to be conveyed.

Subsection “Gene expression analysis of thyroid carcinoma reveals pathways involved in cancer progression and a gene expression signature predictive of disease free survival”: Please rewrite the section title to more accurately describe the data. Consider: "Gene expression analysis of zebrafish thyroid carcinoma reveals pathways involved in disease progression and the same gene signature in human PTC predicts increased disease free survival".

Discussion, first paragraph: please change the word "conserved" to "found" as conserved implies an ancestral relationship.

Discussion, third paragraph: Please acknowledge that A Randomized Phase 2 Study of Single Agent Dabrafenib (BRAFi) vs. Combination Regimen Dabrafenib (BRAFi) and Trametinib (MEKi) in Patients With BRAF Mutated Thyroid Carcinoma. Clinical Trial – NCT01723202 has been open to patient enrollment since 2012.

Discussion, fourth paragraph: – Please clarify how BRAF being able to drive PTCs in adult zebrafish is consistent with BRAF mutations being identified in papillary microcarcinomas. Are BRAF mutations in thyroids observed in benign lesions as they are in nevi? Are no other genetic lesions seen in papillary microcarcinomas – unless these two items are known please rephrase this sentence.

Figure 4 legend: positive is spelled incorrectly.

Regarding Figure 6 in the text, please comment/discuss that RNA-seq data could also be different due to other cell populations being present in the sequenced sample as sorting was not done in this experiment.

11) Figure edits:

Figure 1: Add arrows in merge panel.

Figure 4: Please outline the fish on the fluorescent panel on 4A and 4B so the fluorescent signal on B can be appreciated anatomically.

Panel 6A: Please increase the font size of the red genes – they are too small to read.

Figure 7: Please align the panels.

Figure 7—figure supplement 1: Please define either in to the legend or in the figure that the middle embryos are the ones injected with gRNA and recombinant Cas9 (so the whole embryo is affected). It is hard to tell from the figure the difference between the embryos seen in the middle and embryos on the R side as this figure is currently organized.

---

## [Author Response]

*Essential revisions:*

*1) Given that low degree of detected mutations (19%) and of these only 2 show loss of function (2/46 = 4%) in the cohort sequenced, it is surprising that the twist3 sgRNA causes such a dramatic reversal of phenotype. It is possible that the sequencing method used is underestimating the number of loss of function alleles. Please assess twist protein levels either through western or immunostaining at any time point to confirm significant loss of function of twist3. Alternatively, it the thyroid cells could be sorted at day 5 dpf after imaging and these cells could be sequenced via Next Generation Sequencing to assay the mutation rate. In addition, a negative control gRNA should be injected to ensure that injection of gRNA alone is not affecting the phenotype, especially given the discordance between the alleles predicted to be loss of function and the strong suppression of BRAF-related thyroid dysmorphic changes.*

In the revised manuscript we present new experimental data to address this point. We were not able to identify an antibody that recognizes zebrafish twist3 by immunohistochemistry. Therefore, we performed molecular studies using twist3 sgRNA and an assessment of mutation rate, as suggested.

We utilized a new synthetic sgRNA targeting *twist3* and compared this to a non- targeting sgRNA in embryos injected with rCas9. Utilizing a synthetic sgRNA we find no effect on early embryo development or morphogenesis, in contrast to our initial data using in vitrosynthesized sgRNA. This data prompted us to modify our approach and perform global CRISPR/Cas9 editing using recombinant Cas9, followed by analysis of follicle structure, T4 hormone production, and FACS of thyroid cells followed by a clonal analysis of editing in the *twist3* gene.

A clonal analysis of embryos injected with an sgRNA targeting twist3 shows that 16/21 (76%) clones harbor a range of indels at the targeted site, all predicted to lead to premature stop codon in twist3 (revised Figure 7). We demonstrate that a non-targeting sgRNA produces no indels at the targeted site in 21 clones from 5 embryos (Figure 7), and is also negative by T7 endonuclease assay. CRISPR/Cas9 editing of twist3 suppressed the thyroid phenotype in Tg-BRAF(V600E)-TOM embryos in 32% of larvae. Thus using new reagents we demonstrate that 1) CRISPR editing of twist3 suppresses the BRAF(V600E) phenotype on thyroid follicles, and restores T4 production, 2) the new molecular tools have high specificity for twist3, and 3) a non-targeting sgRNA does not produce molecular alterations in twist3 locus or revert the BRAF(V600E) phenotype. Finally, we performed FACS sorting of thyrocytes, as suggested, and a clonal analysis from these cells revealed indels in twist3 in 1/12 (8%) clones, confirming the presence of a twist3 loss of function mutation in thyrocytes. We acknowledge that the rate of indels in sorted thyrocytes (8%) is lower than observed in globally edited embryos (76%). It is possible that sorting low cell numbers and subsequent PCR amplification introduces bias and leads to an underestimate in the true rate of indels in sorted TOM+ cells. Nevertheless, we do confirm editing at a measurable rate in FACS sorted thyroid cells from embryos with a suppressed phenotype, and no edits were found in TOM+ cells sorted from non-target sgRNA injected animals, so we believe that the claim that we make is reasonable.

*2) The authors have used tissue-specific knockout of twist3 to claim that it is necessary for BRAF^V600E^-induced morphological changes in thyroid structure. Embryos injected with twist3 sgRNA (Figure 7—figure supplement 1) have morphological defects, notably cyclopia, that are reminiscent of general RNA toxicity. The authors should show that the rescue of thyroid structure is not due to non-specific morphological changes caused by RNA toxicity. A variety of control experiments can be performed to exclude this possibility.*

This comment led us to examine alternative sgRNAs to determine if they may be less toxic. Using a synthetic twist3 sgRNA we see no evidence of developmental toxicity, representative embryos at 5dpf are presented in Figure 7—figure supplement 1. Based on this finding we shifted our strategy to perform a global edit of twist3 in the tg-BRAF- TOM transgenic larvae, followed by analysis of follicle structure, hormone production, and FACS isolation of TOM+ thyrocytes from twist3 sgRNA or non-targeting sgRNA injected embryos. These experiments form the basis of a revised Figure 7.

*3) The Abstract asserts that the data demonstrate a genetic requirement for TWIST2 in the earliest step of BRAF^V600E^ transformation. Please rewrite this sentence with a more limited and specific conclusion. Alternatively, if data is already collected regarding whether twist3 CRISPR suppressed onset of BRAF-driven adult thyroid cancer in this model and if existing human sequencing data suggest this could be an early event in PTC development, please add these two lines of data.*

We have edited the sentence in the Abstract to present a more limited and specific conclusion. The sentence now reads, “These data suggest that expression of TWIST2 plays a role in an early step of BRAF^V600E^-mediated transformation.”

*4) There are instances where the authors should fortify the claims they are making. First, in the Abstract it is stated that there is a 'genetic requirement for TWIST2 in the earliest step of BRAF^V600E^-mediated transformation'. While twist3 might be important for BRAF^V600E^-driven changes in thyroid morphology, it is not clear that these morphological changes represent an early stage or step of oncogenic transformation. It would bolster their case if the authors show an effect of twist3 loss on BRAF^V600E^-driven tumorigenesis. Can they show that animals with thyroid-specific loss of twist3 also have delayed tumor onset? If not, then the authors' claims should be rephrased. Second, the sentence 'We identify an orthologue of human TWIST2, twist3, as a key mediator of BRAF^V600E^ induced EMT in thyrocytes' has similar issues. While an EMT signature (among others) is seen in BRAF^V600E^-expressing thyrocytes, the authors should do more before unequivocally stating this is EMT. Showing changes in expression of canonical EMT markers (e.g. E cadherin, N cadherin) is necessary.*

We have rephrased the sentence in the Abstract as noted above to present a more limited conclusion. Experiments to address the effects of twist3 loss of function on tumor development are ongoing and beyond the scope of the current manuscript. We agree that the connection between the early morphologic phenotypes and long term effects on tumor development is an important topic for future study.

We present new data to demonstrate a loss of E-cadherin staining in BRAF(V600E) expressing thyrocytes, consistent with EMT. This data is presented in Figure 3—figure supplement 1.

*5) All three reviewers also agree that the authors have not done the experiments to be able to conclude that there is a "pivotal functional role for TWIST expression during the earliest stage of malignant transformation by oncogenic BRAF". The conclusions should be softened in the text. The work would form the basis of a nice new paper. To support this conclusion the authors would need to do a variety of human sequencing and cell culture work. We feel that all these experiments are clearly beyond the two-month time limit recommended by eLife. For example, sequencing of human PTCs for early events would be a whole other collaboration/project and cell culture work may not recapitulate the indolent nature of human PTC modeled in this paper and could take a long time, may up to a year. Given the time restraints suggested by eLife, we are comfortable instead having the authors remove the sentence at the end of the Discussion.*

We have removed the sentence highlighted by the reviewers.

*Non-essential revisions:*

*1) One deficiency of the current manuscript is lack of human disease equivalent for the presented results. If available, please add data using human samples of thyroid cancer or thyroid cancer cell lines that this pathway is important, conserved and relevant to human thyroid cancer driven by BRAF^V600E^.*

In our manuscript we highlighted work showing that BRAF promotes invasion of thyroid cancer cells through an EMT and Snail dependent mechanism (Baquero, et al). We also discuss work demonstrating that TWIST is associated with lymph node metastasis in papillary thyroid carcinoma (Wang, et al), and that it modulates phenotypes in anaplastic thyroid cancer (Salerno et al).

In order to add more context to the significance of EMT in thyroid cancer we have added a reference (Vasko et al.) to a key publication that provided molecular evidence for an upregulation of EMT effectors at the invasive front in human papillary thyroid cancers.

*2) The authors show that the effect of BRAF^V600E^ in thyroid morphogenesis can be rescued by combined treatment with BRAF and MEK inhibitors but not by either of them individually. The authors do not explain any further why this happens. Have the authors tried higher concentrations with individual inhibitors to see if they can show rescue?*

The data we present represent the highest concentration of the inhibitors that is consistent with normal development. Higher concentrations resulted in severe developmental abnormalities that precluded analysis of effects on thyroid structure.

*3) This is not necessarily a 'concern', but could the authors determine whether their zebrafish PTC signature aligns more closely with human (or mouse) PTC versus PDTC. If the zebrafish signature is more similar to PTCs, then it would help their argument that they are modeling a more indolent form of the disease.*

We agree that a comparison of PTC vs. PDTC may be informative. Unfortunately, the sample size of the publicly available PDTC gene expression datasets (n=17 PDTC samples in GEO GSE76039), makes it very challenging to perform a meaningful comparison.

*4) The western blots in Figure 4 should be quantified as the changes are subtle.*

As suggested we performed this analysis and include it in a revised Figure 4.

*5) The histology images in Figure 5 are difficult to see. Zoomed in insets of certain important regions/features within the PTC will help in visualizing these.*

As suggested we have provided zoomed insets to better visualize key histopathologic features, such as skeletal muscle invasion.

*6) The chromatograms in Figure 7 and Figure 7—figure supplement 1 show wide peaks in mutated regions of the twist3 gene. In the clonal analyses described, where only one copy of the gene is being sequenced, regular-width peaks along with a commensurate frameshift downstream of the indel would be expected. It is not clear why the chromatograms look like they do.*

In the revised manuscript these specific chromatograms have been removed. A new chromatogram included in Figure 7 shows a 3 nucleotide indel, and the peak width is unchanged in the deleted region.

*7) Please describe in the text how was the gene signature used in the data presented in Figure 6 chosen. Did this gene signature predict improved OS for patients with PTC? If not, please hypothesize regarding any discrepancy.*

As we described in the Materials and methods section and the Figure 6 legend, we identified the top differentially expressed genes using standard metrics (FDR ≤ 1x10^-6^, FC ≤1.5). This resulted in a very stringent selection of 58 genes differentially regulated in BRAF thyroid tumors compared to normal thyroid tissue. In order to test whether these genes stratify a clinically meaningful outcome, such as disease free survival (DFS), we utilized RNAseq data from the TCGA papillary thyroid cancer cohort. As described in Materials and methods we calculated a rank per subject for the signature in the TCGA cohort. We then stratified subjects by median rank and performed an analysis of DFS. Almost no deaths occurred in the TCGA PTC cohort, which is typical for PTC, so an analysis of overall survival is not possible. We edited the Materials and methods section to improve clarity.

*8) Please comment in the text or Discussion why in Figure 2 selemetinib decreases number of follicles significantly.*

We have commented in the text that MAPK inhibition results in a modest decrease in follicle number, consistent with a proliferative requirement in follicle morphogenesis.

*9) Figure 4 and Figure 5 and the text associated with them would be clarified by addressing the following items.*

*a) What is the significance if the pathologic features shown in Figure 5 in human PTC if any?*

We have edited the text to note that the features highlighted in Figure 5 are diagnostic of thyroid carcinoma.

*b) Please describe in the text whether the one-year animals get metastases.*

We did not perform an exhaustive histologic study for distant metastatic disease instead focusing on local invasion. In fact, it is difficult to exclude metastasis as a mechanism for some of the disease we see in soft tissue and skeletal muscle.

*c) Were the animals with the desmoplastic reaction the same animals with the other pathologic features?*

These animals were distinct.

*d) If the animals look different at 12m with the tumors as compared to 5 mpf, can a photo of 12mpf fish with thyroid cancer be included at the beginning of Figure 5?*

The animals have a similar external appearance at 5mpf and 12mpf.

*e) Please also include more up close photos of the cytologic features in Figure 5, especially of the nuclear grooves and elongated nuclei so these can be appreciated.*

We improved the resolution of this figure to allow better appreciation of these features.

*f) Please also annotate the structures being invaded in histologic structures in Figure 4 and Figure 5 to ensure non-zebrafish audience can appreciate the data.*

We have provided insets to highlight skeletal muscle invasion.

*10) The manuscript would benefit from several small edits in order to clarify the work. Most of these edits will help the work be more accessible to a non-zebrafish audience.*

*Text edits:*

*Abstract: Please clarify that developmental effects of BRAF^V600E^-driven changes in thyroids were reversed with drug treatment.*

We edited the Abstract to address this point.

*Introduction, second paragraph: Please define the acronym NIS.*

We edited the text to address this point.

*Subsection “BRAF^V600E^ expression leads to TGF-β and EMT gene expression changes in thyrocytes”: Please name the tumors that the cell lines were derived from (thyroid?) and clarify how an ERK signature is derived by inhibiting MEK. Alternatively, simplify the sentence.*

The ERK signature we utilize was derived from an analysis of BRAF(V600E) mutant melanoma and colon cancer cell lines (Pratilas, et al). In the Pratilas manuscript the authors examine genes that change in expression 8h after treatment with a MEK inhibitor. We have simplified the sentence in the text to avoid confusion.

*Subsection “BRAF^V600E^ expression leads to TGF-β and EMT gene expression changes in thyrocytes”: Please clarify which of these groups of upregulated genes were also seen in the human data.*

GSEA analysis presents the genes that are most associated with a positive enrichment as hash marks on the left side of the plot. There is variability in how GSEA results are presented in the scientific literature. We do not feel that a complete listing of the GSEA results would add significantly to the manuscript.

*Subsection “BRAF^V600E^ expression in thyrocytes induces thyroid cancer in adult zebrafish”: “We analyzed a second cohort of tg-BRAF^V600E^-TOM adult animals at 12 mpf and found evidence of progressive disease and histologic and cytological features consistent with thyroid carcinoma (Figure 5).”: This sentence is redundant with the following sentence, please remove or combine with the following sentence. Or clarify the point being made.*

We edited the text to address this point, the sentences were combined.

*Subsection “BRAF^V600E^ expression in thyrocytes induces thyroid cancer in adult zebrafish”: The last sentence is also redundant, please remove this sentence or clarify if a different point was meant to be conveyed.*

We edited an earlier sentence to avoid redundancy.

*Subsection “Gene expression analysis of thyroid carcinoma reveals pathways involved in cancer progression and a gene expression signature predictive of disease free survival”: Please rewrite the section title to more accurately describe the data. Consider: "Gene expression analysis of zebrafish thyroid carcinoma reveals pathways involved in disease progression and the same gene signature in human PTC predicts increased disease free survival".*

We have edited the section title to address this point. The revised title reads: “Gene expression analysis of zebrafish thyroid carcinoma reveals pathways involved in disease progression and a gene expression signature that is predictive of disease free survival in human PTC”.

*Discussion, first paragraph: please change the word "conserved" to "found" as conserved implies an ancestral relationship.*

We edited the text to address this point.

*Discussion, third paragraph: Please acknowledge that A Randomized Phase 2 Study of Single Agent Dabrafenib (BRAFi) vs. Combination Regimen Dabrafenib (BRAFi) and Trametinib (MEKi) in Patients With BRAF Mutated Thyroid Carcinoma. Clinical Trial – NCT01723202 has been open to patient enrollment since 2012.*

We acknowledge this trial in the manuscript as suggested.

*Discussion, fourth paragraph: Please clarify how BRAF being able to drive PTCs in adult zebrafish is consistent with BRAF mutations being identified in papillary microcarcinomas. Are BRAF mutations in thyroids observed in benign lesions as they are in nevi? Are no other genetic lesions seen in papillary microcarcinomas – unless these two items are known please rephrase this sentence.*

There are no known benign lesions harboring a BRAF(V600E) mutation in thyroid cancer, so the presence of a BRAF(V600E) mutation even in a microscopic specimen is diagnostic of cancer. BRAF(V600E) mutations are the most common somatic alteration in papillary thyroid cancer. For this reason, we feel that the comparison to papillary microcarcinoma is reasonable.

*Figure 4 legend: positive is spelled incorrectly.*

We corrected this misspelling.

*Regarding Figure 6 in the text, please comment/discuss that RNA-seq data could also be different due to other cell populations being present in the sequenced sample as sorting was not done in this experiment.*

We feel that the manuscript contains sufficient methodologic detail as we state that Tomato+ tissue was microdissected. While we cannot exclude the possibility that there is contaminating tissue, the gross specimens submitted were homogenous by examination under a dissecting scope using epifluorescence, suggesting to us to that the tissue is almost entirely thyroid in origin.

*11) Figure edits:*

*Figure 1: Add arrows in merge panel.*

The figure has been edited to address this point.

*Figure 4: Please outline the fish on the fluorescent panel on 4A and 4B so the fluorescent signal on B can be appreciated anatomically.*

We adjusted the contrast to better allow visualization of the location of the TOM+ thyroid in the ventral jaw and added an arrow to identify the site of fluorescence.

*Panel 6A: Please increase the font size of the red genes – they are too small to read.*

Figure 6 has no gene names called out, so we are uncertain of the query.

*Figure 7: Please align the panels.*

This figure has been revised.

*Figure 7—figure supplement 1: Please define either in to the legend or in the figure that the middle embryos are the ones injected with gRNA and recombinant Cas9 (so the whole embryo is affected). It is hard to tell from the figure the difference between the embryos seen in the middle and embryos on the R side as this figure is currently organized.*

This figure has been revised.